# MYL3 protects chondrocytes from senescence by inhibiting clathrin-mediated endocytosis and activating of Notch signaling

He Cao [1,2,6], Panpan Yang[1,2,6], Jia Liu[3,6], Yan Shao[1,2], Honghao Li [1,2], Pinglin Lai[1,2], Hong Wang[1,2], Anling Liu[4], Bin Guo[4], Yujin Tang [3], Xiaochun Bai [1,4] ✉ & Kai Li [1,2,5] ✉

As the unique cell type in articular cartilage, chondrocyte senescence is a crucial cellular event contributing to osteoarthritis development. Here we show that clathrin-mediated endocytosis and activation of Notch signaling promotes chondrocyte senescence and osteoarthritis development, which is negatively regulated by myosin light chain 3. Myosin light chain 3 (MYL3) protein levels decline sharply in senescent chondrocytes of cartilages from model mice and osteoarthritis (OA) patients. Conditional deletion of Myl3 in chondrocytes significantly promoted, whereas intra-articular injection of adeno-associated virus overexpressing MYL3 delayed, OA progression in male mice. MYL3 deficiency led to enhanced clathrin-mediated endocytosis by promoting the interaction between myosin VI and clathrin, further inducing the internalization of Notch and resulting in activation of Notch signaling in chondrocytes. Pharmacologic blockade of clathrin-mediated endocytosis-Notch signaling prevented MYL3 loss-induced chondrocyte senescence and alleviated OA progression in male mice. Our results establish a previously unknown mechanism essential for cellular senescence and provide a potential therapeutic direction for OA.

Cellular senescence is a state of irreversible growth arrest, which has been noted as one of the hallmarks of aging[1,2]. The typical features of senescent cells include elevated expression of p21-, p16[INK4A]-, and p53-positive staining for senescence-associated β-galactosidase, and secretion of senescence-associated secretory phenotype (SASP) factors[3,4]. Osteoarthritis (OA) is the most prevalent chronic and disabling joint disorder with multiple risk factors including age, obesity, trauma, genetics, and sex, which aging is the most prominent one[5,6]. Chondrocytes, a unique cell type in cartilage tissue, undergo cellular senescence with aging and contribute to dysregulation of anabolic and catabolic homeostasis in cartilage during OA development[7-9]. Recent studies revealed that the removal of senescent

[1]Academy of Orthopedics, Guangdong Province, Guangdong Provincial Key Laboratory of Bone and Joint Degeneration Diseases, The Third Affiliated Hospital of Southern Medical University, Guangzhou, China. [2]The Third School of Clinical Medicine, Southern Medical University, Guangzhou, China. [3]Guangxi Key Laboratory of basic and translational research of Bone and Joint Degenerative Diseases, Guangxi Biomedical Materials Engineering Research Center for Bone and Joint Degenerative Diseases, Department of Orthopedics, Affiliated Hospital of Youjiang Medical University for Nationalities, Baise, Guangxi, China. [4]State Key Laboratory of Organ Failure Research, Department of Cell Biology, School of Basic Medical Sciences, Southern Medical University, Guangzhou, China. [5]Guangzhou Key Laboratory of Neuropathic Pain Mechanism at Spinal Cord Level, The Third Affiliated Hospital of Southern Medical University, Guangzhou, China. [6]These authors contributed equally: He Cao, Panpan Yang, Jia Liu. ✉e-mail: baixc15@smu.edu.cn; lk516433415@smu.edu.cn

chondrocytes using transgenic models or pharmaceutical intervention would decrease SASP secretion and attenuate posttraumatic and aged-OA progression in mice, indicating that targeting senescent cells (SnCs) could be a promising approach for OA therapy[10–12]. However, the occurrence and regulatory mechanism of chondrocyte senescence has not yet been fully addressed.

Recent reports have elaborated that disrupted endocytosis including clathrin-mediated endocytosis (CME), the major endocytic pathway in eukaryotic cells, is involved in cellular senescence[13–15]. CME controls the transport of the majority of transmembrane receptors and transporters from the plasma membrane to the cytoplasm with the aid of a clathrin coat[16,17]. Although this transport is essential for numerous cellular processes, such as the maintenance of membrane homeostasis and the regulation of intercellular signaling[18], the role of CME in chondrocyte senescence and OA development has not been reported.

The myosins are a large family of actin-activated mechanoenzymes that generate force and movement along actin filaments by hydrolyzing ATP to produce mechanical energy[19,20]. Previous studies have identified several myosins as specific binding partners of clathrin that associate with clathrin-coated structures (CCS), among which myosin VI (MYO6) serves as a motor protein to move CCS into the cell by moving towards the minus ends of actin filaments[21–23].

In this study, we uncovered the vital role of myosin light chain 3 (MYL3) in CME-mediated Notch signaling activation and chondrocyte senescence. Loss of MYL3 in chondrocytes leads to enhanced CME by promoting the interaction between MYO6 and clathrin, and consequently induces the internalization of Notch receptors and nuclear translocation of Notch intracellular domain (NICD), resulting in the activation of Notch signaling, chondrocyte senescence and OA.

## Results

### CME is enhanced and MYL3 expression is decreased in senescent chondrocytes

To investigate the role of CME in chondrocyte senescence, we examined endocytosis of Alexa Fluor 594-conjugated transferrin as a marker of general CME. Primary chondrocytes at passage seven or with $H_2O_2$ treatment displayed senescence phenotypes with increased senescence-associated β-galactosidase (SA-β-Gal) positivity and p16[INK4a] expression (Fig. 1a and Supplementary Fig. 1a). We also noticed a stronger transferrin uptake and enhanced biotinylated transferrin uptake (Fig. 1b and Supplementary Fig. 1b, c), indicating that CME is enhanced in senescent chondrocytes.

Further, to explore the mechanisms that control CME and cellular senescence in chondrocytes, we performed proteome-wide screening with articular cartilage from young (2-month-old) and aged (12-month-old) mice. In the 3171 proteins identified, MYL3, a member belonging to the myosin families which known to be involved in the regulation of CME, was found to be the most downregulated protein among the myosin families in the cartilage of aged mice compared to young mice. (Fig. 1c and Supplementary Table 1). Next, immunofluorescence (IF) staining of MYL3 was performed in articular cartilage from mice aged 2, 12, and 24 months, and we confirmed that as cartilage degeneration progressed, Osteoarthritis Research Society International (OARSI) scores, markers for senescence (p16[INK4a]), DNA damage (γ-H2AX), and CME (MYO6 and clathrin) in articular chondrocytes were all upregulated. However, the ratio of MYL3-positive chondrocytes in aged mice was significantly lower compared with that in young mice (Fig. 1d, e, Supplementary Fig. 1d). In addition, we confirmed the reduced expression of MYL3 in damaged cartilage compared to intact human cartilages, accomplished with increased senescence, DNA damage, and CME markers (Fig. 1f, g, Supplementary Fig. 1e). Additionally, in the surgical destabilization of the medial meniscus (DMM) surgery-induced posttraumatic OA mice model, we also detected

significantly decreased MYL3 level with enhanced senescence and CME in chondrocytes of articular cartilage along with the progression of OA (Fig. 1h, i, Supplementary Fig. 1f). Western blot analysis showed that chondrocytes at passage seven or chondrocytes exposed to $H_2O_2$ displayed increased protein levels of senescence markers and decreased protein levels of MYL3 (Supplementary Fig. 1g, h).

Taken together, these data demonstrated that CME is enhanced in senescent chondrocytes, while MYL3 is reduced during chondrocyte senescence and OA progression.

### MYL3 inhibits CME and protects chondrocytes against cellular senescence

We then explored the role of MYL3 in CME and cellular senescence in chondrocytes. Firstly, we traced CME by co-staining MYO6 and clathrin in chondrocytes at different passages and observed elevated CME at passage seven compared with passage two (Supplementary Fig. 2a). Inhibition of CME process through pharmacological (dynasore) or genetic (clathrin knockdown or enforced DynK44A mutant expression) approaches decreased the levels of senescence markers at different passages or with $H_2O_2$ stimulation (Supplementary Fig. 2b, c). These data suggest that inhibition of CME process protect against senescent in chondrocytes. Next, in chondrocytes with *Myl3* knockdown lentivirus (*Myl3*-KD), we noticed a stronger transferrin uptake than control, while overexpression of *Myl3* showed less (Fig. 2a). Consistent with the results of $H_2O_2$ stimulation, knockdown of *Myl3* enhanced, while overexpression of *Myl3* (*Myl3*-OE) inhibited, transferrin uptake (Supplementary Fig. 3a). To illustrate how MYL3 regulated CME, we tested whether MYL3 interacted with MYO6 or clathrin, two major regulators of CME. Co-immunoprecipitation analysis revealed that MYL3 interacted with MYO6, but did not bind to clathrin (Fig. 2b, c). In chondrocytes stimulated by $H_2O_2$, the interaction, and co-localization of MYO6 and clathrin were enhanced compared with controls (Supplementary Fig. 3b). Moreover, following *Myl3* knockdown, we noted that the interaction between MYO6 and clathrin was strengthened but was restrained when MYL3 was overexpressed (Fig. 2d, Supplementary Fig. 3b). Using IF double staining, we confirmed that in chondrocytes of passage seven or with $H_2O_2$ stimulation, co-located MYO6 and clathrin were enhanced by *Myl3* knockdown and decreased when MYL3 was overexpressed (Fig. 2e, Supplementary Fig. 3c). We noticed that knockdown of *Myl3* enhanced SA-β-Gal activity and the protein level of senescence markers and SASP factor (Mmp13) at both passages two and seven, while overexpression of *Myl3* showed opposite effects (Fig. 2f–i). Moreover, following treatment with $H_2O_2$, knockdown of *Myl3* also significantly increased SA-β-Gal activity and elevated the protein level of senescence markers and SASP factor, while overexpression of *Myl3* exerted the opposite effects (Supplementary Fig. 3d–f). Taken together, our data indicate that MYL3 negatively regulates CME to protect against cellular senescence in chondrocytes by inhibiting the interaction between MYO6 and clathrin.

### Deletion of MYL3 in chondrocytes promotes chondrocyte senescence and OA progression in mice

Next, we generated *Col2a1-Cre: Myl3fl/fl* (*Myl3*-KO) conditional knockout mice and *Col2a1-CreER^{T2}:Myl3fl/fl* (*Myl3*-iKO) inducible conditional knockout mice. The *Myl3*-KO mice exhibited similar gross appearance and morphology at embryonic day 18.5 and at 3 months of age (Supplementary Fig. 4a–d), while the organization of the growth plates and body weights were also comparable with those of control mice (Supplementary Fig. 4e–h). These data indicated that deletion of *Myl3* in chondrocytes did not affect the skeletal development of the mice.

Next, the histopathological changes of joints in aged *Myl3*-KO mice were analyzed (Fig. 3a). IHC analysis confirmed the knockout efficiency with decreased expression in chondrocytes from 6- and 18-

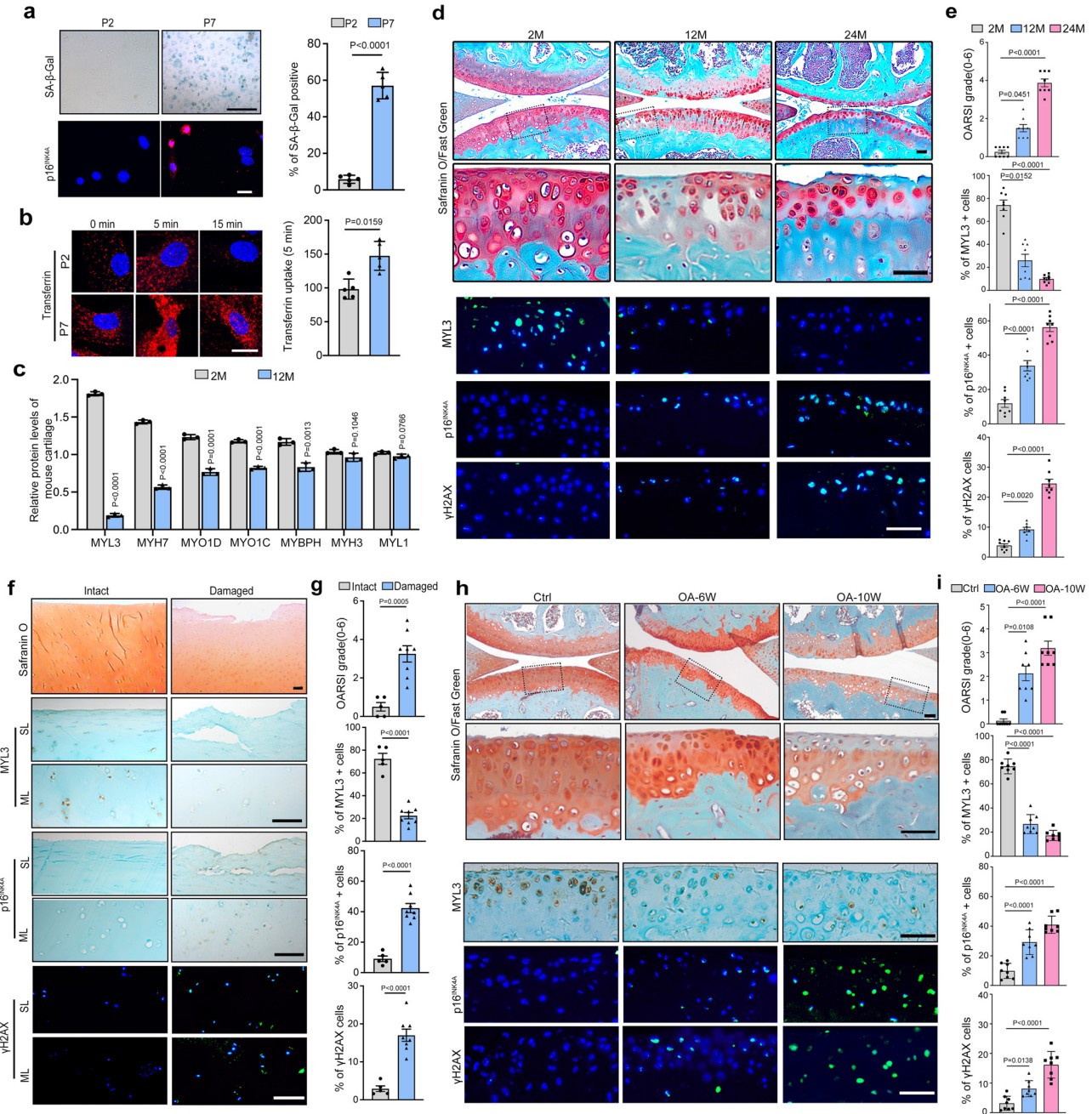

**Fig. 1 | CME is enhanced and MYL3 expression is decreased in senescent chondrocytes. a** Images and quantification of SA-β-Gal positivity and IF staining of p16[INK4A] in mouse primary chondrocytes at passage 2 and 7. *n* = 5, bars = 25 µm. **b** Images and quantification of transferrin endocytosis in mouse primary chondrocytes at passages 2 and 7. *n* = 5, bars = 25 µm. **c** Quantification of the relative expression of myosins with a proteome-wide screen for the proteins of articular cartilage from 2- or 12-month-old mice. **d** Images of Safranin-O and IF staining of MYL3, p16[INK4A], γH2AX and quantification (**e**) of OARSI scores, MYL3-, p16[INK4A]- and γH2AX-positive cells and in articular cartilage from 2-, 12- and 24-month-old mice, blue indicates DAPI staining of nuclei. *n* = 8, bars = 50 µm. **f** Images of Safranin-O, IHC of MYL3 and p16[INK4A], and IF staining of γH2AX and quantification of OARSI,

MYL3-, p16[INK4A]- and γH2AX-positive cells (**g**) in intact (*n* = 5) and damaged (*n* = 9) articular cartilage sections collected from OA patients. Bars = 50 µm. **h** Images of Safranin-O staining, IHC staining for MYL3 and IF staining of p16[INK4A] and γH2AX, and quantification of OARSI scores and MYL3-, p16[INK4A]- and γH2AX-positive cells (**i**) in articular cartilage from mice of control, 6- and 10- weeks post DMM surgery. *n* = 8, bars = 50 µm. Data are all shown as means ± SD. *P* values are from two-tailed Mann–Whitney *U* test (**b**), two-tailed unpaired *t* test (**a**, **g**), one-way ANOVA test followed by Tukey's post hoc test (p16[INK4A], γH2AX in **e**, MYL3, p16[INK4A], γH2AX in **i** and Kruskal–Wallis test followed by Dunn's post hoc test (OARSI, MYL3 in **e**, OARSI in **i**). *n* indicates the number of biologically independent samples, mice per group, or human specimens. Source data are provided as a Source Data file.

month-old *Myl3*-KO mice (Fig. 3b). We found that the cartilage destruction in 6-month-old *Myl3*-KO mice was worse, and became more severe at 18 months of age, accompanied with aggravated synovial inflammation but comparable osteophyte maturity compared with age-matched control mice (Fig. 3c and Supplementary Fig. 5a). Accordingly, the OARSI scores, as well as synovial inflammation scores

were both significantly increased in aged *Myl3*-KO mice (Fig. 3d). IF and IHC analysis confirmed the dramatically increased expression of senescence (p16[INK4a]), DNA damage (γ-H2AX), and catabolic marker (MMP13), as well as decreased nuclear level of HMGB1 in the cartilage of aged *Myl3*-KO mice (Fig. 3e and Supplementary Fig. 5b). Subsequently, male *Myl3*-iKO mice and control littermates were used to

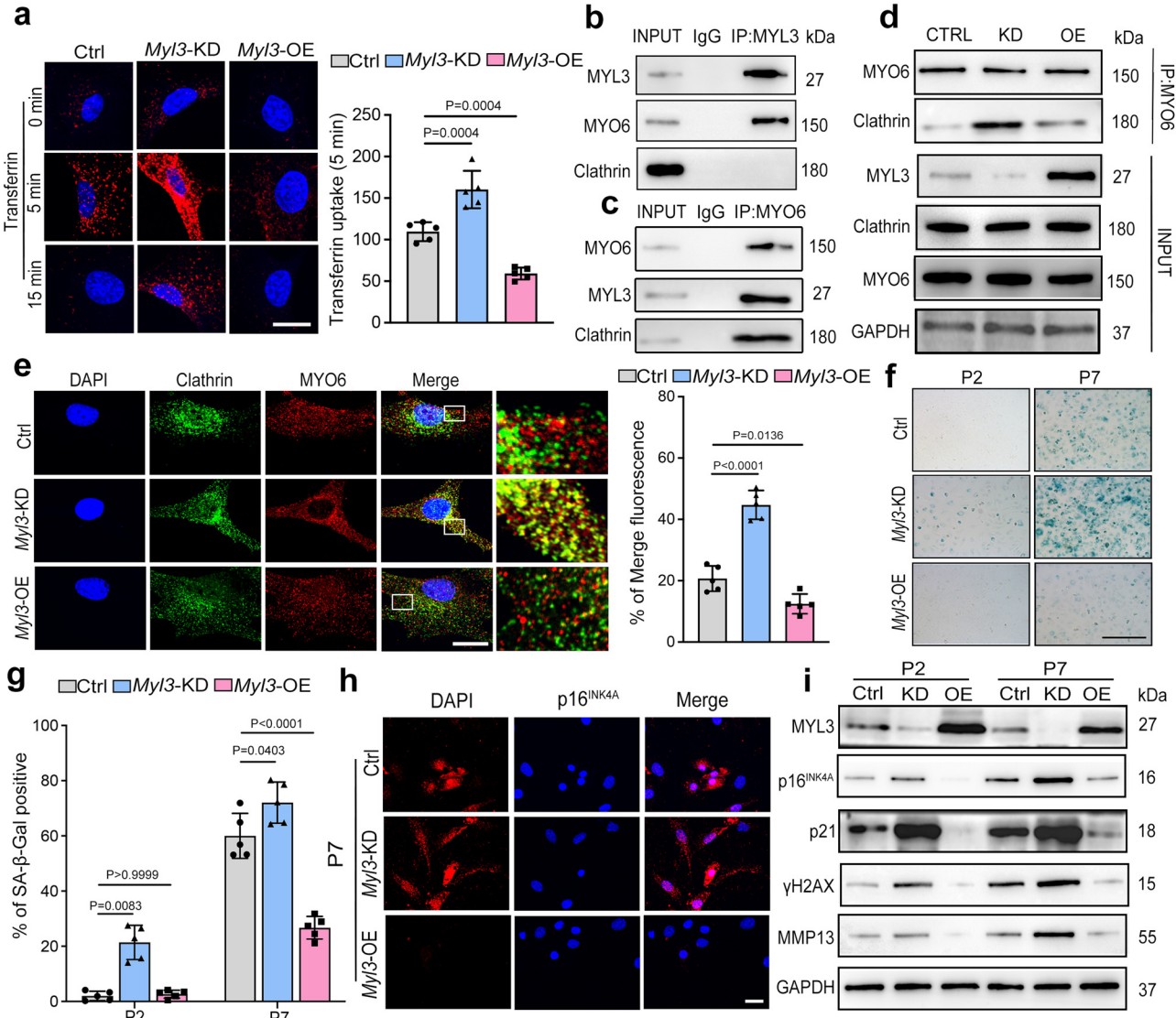

**Fig. 2 | MYL3 negatively regulates CME and cellular senescence in chondrocytes. a** Images and quantification of endocytosed transferrin in mouse primary chondrocytes transfected with control, *Myl3* shRNA knockdown (*Myl3*-KD) or overexpression adenovirus particles (*Myl3*-OE) at passage 4. *n* = 5, bars = 25 μm. **b** IP using control IgG and anti-MYL3 antibody in mouse primary chondrocytes to detect the binding of MYL3 with MYO6 and Clathrin. **c** IP revealed the binding of MYO6 and MYL3 or Clathrin in chondrocytes. **d** IP revealed the binding of MYO6 and Clathrin in mouse primary chondrocytes transfected with control, *Myl3*-KD, or *Myl3*-OE. **e** IF images and quantification of Clathrin (green) and MYO6 (red)

co-localization at passage 7, SA-β-Gal positivity at passage 2 or 7 (**f, g**) in mouse primary chondrocytes transduced with control, *Myl3*-KD or *Myl3*-OE. *n* = 5, bars = 25 μm. **h** Images of p16INK4A IF and protein levels of MYL3, p16INK4A, P21, γH2AX, and MMP13 (**i**) in mouse primary chondrocytes transfected with control, *Myl3*-KD or *Myl3*-OE. Bars = 25 μm. Data are representative of three independent experiments and are all shown as means ± SD. *P* values are from a one-way ANOVA test followed by Tukey's post hoc test (**a, e, g**). *n* indicates the number of biologically independent samples or mice per group. Source data are provided as a Source Data file.

examine the role of *Myl3* deletion in DMM surgery-induced OA (Fig. 3f). IHC analysis confirmed the knockout efficiency in chondrocytes from *Myl3*-iKO mice after tamoxifen injection (Fig. 3g). Unsurprisingly, histological examination showed more severe OA progression at both 6- and 10 weeks post DMM surgery in *Myl3*-iKO mice, as evidenced by reduced cartilage areas, aggravated synovial inflammation, partially-increased osteophyte maturity, and markedly increased OARSI scores (Fig. 3h, i and Supplementary Fig. 5c). IHC analysis demonstrated enhanced cellular senescence with increased senescence marker, DNA damage, and catabolic marker, as well as the decreased nuclear level of HMGB1 in cartilage from *Myl3*-iKO mice compared with control mice at 6- and 10 weeks post DMM surgery (Fig. 3j and Supplementary Fig. 5d). Therefore, these results reveal that *Myl3* deletion resulted in accelerated OA development by promoting chondrocyte senescence.

## Overexpression of MYL3 attenuates development of age-related and experimental OA in mice

To explore the role of upregulated MYL3 in chondrocytes and in OA pathogenesis, adeno-associated virus encoding *Myl3* overexpression plasmid was intra-articularly injected into wild-type mice (*Myl3*-OE). The histopathological changes of joints in *Myl3*-OE mice were then analyzed at 6 months and 18 months of age (Fig. 4a). IHC analysis showed significantly elevated MYL3 expression in the articular cartilage of *Myl3*-OE mice compared with controls (Fig. 4b). Histological examination showed that the cartilage destruction and proteoglycan loss were significantly ameliorated, while OARSI scores were significantly lower in aged *Myl3*-OE mice compared with controls, while the synovial inflammation and osteophyte maturity were not changed (Fig. 4c, d and Supplementary Fig. 6a). IHC detection showed

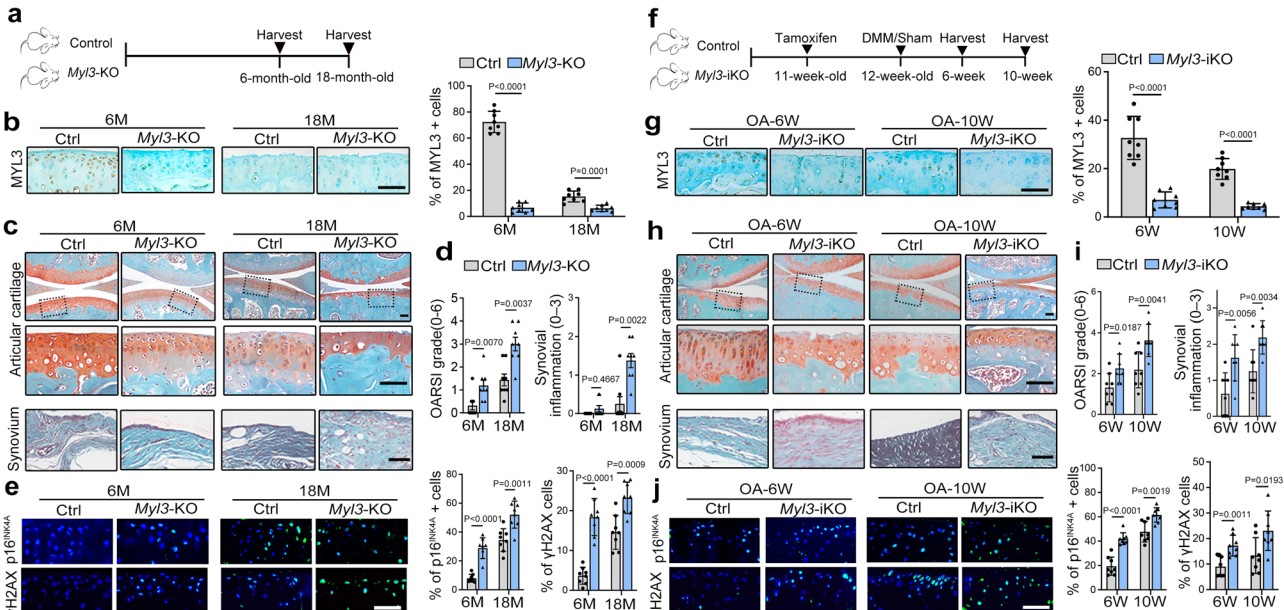

**Fig. 3 | Deletion of MYL3 in chondrocytes promotes chondrocytes senescence and OA progression in mice. a** Schematic illustration of the spontaneous age-related OA model in control and *Myl3*-KO mice. **b** Images and quantification of IHC staining of MYL3, Safranin-O staining images (**c**), OARSI and synovial inflammation scores (**d**), and IF staining images of p16^INK4A and γH2AX (**e**) in sagittal sections of joints from control and *Myl3*-KO mice at 6 months and 18 months. *n* = 8 independent biological replicates per group, bars = 50 μm. **f** Schematic illustration of posttraumatic OA model with DMM surgery in control and *Myl3*-iKO mice. **g** Images and quantification of IHC staining of MYL3, Safranin-O staining images (**h**), OARSI and synovial inflammation scores (**i**), and IF staining images of p16^INK4A and γH2AX (**j**) in sagittal sections of joints from control and *Myl3*-iKO mice at 6- and 10- weeks post-surgery. *n* = 8 independent biological replicates per group, bars = 50 μm. Data are all shown as means ± SD. *P* values are from two-tailed Mann–Whitney *U* test (OARSI, synovial inflammation scores in **d**) and two-tailed unpaired *t* test (remaining quantification). *n* indicates the number of biologically independent samples or mice per group. Source data are provided as a Source Data file.

decreased senescence, DNA damage, and catabolic marker, as well as increased nuclear level of HMGB1 expression in chondrocytes of the *Myl3*-OE group (Fig. 4e and Supplementary Fig. 6b). We also analyzed the influence of MYL3 overexpression on posttraumatic OA (Fig. 4f). Consistently, increased protein abundances of MYL3 were confirmed in the articular cartilage of *Myl3*-OE mice (Fig. 4g). Amelioration of DMM-induced cartilage breakage was observed at both 6- and 10 weeks post-surgery, but synovial inflammation was only improved at 10 weeks post-surgery (Fig. 4h, i and Supplementary Fig. 6c). Furthermore, IHC analysis showed decreased senescence, DNA damage, and catabolic marker, as well as increased nuclear level of HMGB1 expression in *Myl3*-OE mouse cartilage after DMM surgery (Fig. 4j and Supplementary Fig. 6d). Thus, our data suggest that overexpression of MYL3 ameliorates age-related and posttraumatic OA manifestations in mice through suppression of chondrocyte senescence.

## Loss of MYL3 promotes the activation of Notch signaling and induces senescence by enhancing CME

To comprehensively discuss the underlying mechanisms through which MYL3 regulates chondrocyte senescence, we performed RNA sequencing analysis with primary chondrocytes from *Myl3*-KO and control mice. Among the 142 differentially expressed genes, 99 genes were upregulated and 43 genes were downregulated in *Myl3*-KO chondrocytes versus controls. The representative top 20 most significantly upregulated genes with altered expression are shown in Supplementary Table 2. From these we focused on the Notch transcription factor *Hes1*, which was upregulated 4.8-fold (Fig. 5a). Furthermore, KEGG pathway analysis revealed an enrichment in differentially-regulated genes associated with Notch signaling (Supplementary Table 3, Fig. 5b). To validate these results, qRT-PCR and western blot analysis were performed and the results confirmed that *Myl3* deletion enhanced expression of Notch-related (HES1, HEY1 and NICD) and senescence-related genes (p16, p53) (Fig. 5c, d). IF staining confirmed the enhancement of NICD in the DMM model, age-related

OA, and damaged cartilage from OA patients (Supplementary Fig. 7a–c), suggesting that Notch signaling is activated in senescent chondrocytes in vivo.

Genetic studies suggest that activation of Notch signaling requires CME[24], so we then questioned whether MYL3 regulated Notch signaling while blocking CME. First, we confirmed that Notch signaling is activated via CME in senescent chondrocytes, as the co-localization of NICD and clathrin were enhanced following treatment with jag1 recombinant protein (Fig. 5e). We also found that the co-localization of NICD with clathrin was enhanced in senescent chondrocytes or in cartilage from *Myl3*-KO mice (Supplementary Fig. 7d, e), confirming that Notch signaling is activated in senescent chondrocytes via CME process in vivo. The enhanced expression and co-localization of NICD with clathrin, RAB5, or RAB7 caused by jag1 treatment could be blocked by treatment with the Notch pathway inhibitor RO4929097, dynasore or MYL3 overexpression (Fig. 5e, Supplementary Fig. 8a, b). Compared with control, *Myl3*-KO chondrocytes had enhanced expression of NICD and its co-localization with Clathrin, RAB5, or RAB7, which could be eliminated via treating with RO4929097 or CME inhibitor dynasore, DynK44A overexpression or clathrin knockdown (Fig. 5f, Supplementary Fig. 9a–c). Western blot analysis showed that expression of p16^INK4a, p53, γH2AX, HES1, and HEY1 were all upregulated with jag1 treatment but downregulated in jag1 with RO4929097, dynasore, and MYL3 overexpression treatment (Fig. 5g). Upregulated expression of p16^INK4a, p53, γH2AX, HES1, and HEY1 caused by *Myl3* knockout could be recovered with RO4929097, dynasore, DynK44A overexpression or clathrin knockdown (Fig. 5h). Moreover, we observed that the constitutively active form of Notch (extracellular domain-deprived Notch mutant, Notch-DeltaE) could upregulate the expression of p16^INK4a, p53, γH2AX, HES1, and HEY1, which could be reversed by RO4929097, MYL3 overexpression, dynasore, and clathrin knockdown (Fig. 5i).

Further, we noticed that jag1 treatment or *Myl3*-KO promoted the senescence of chondrocytes with enhanced SA-β-Gal activities and

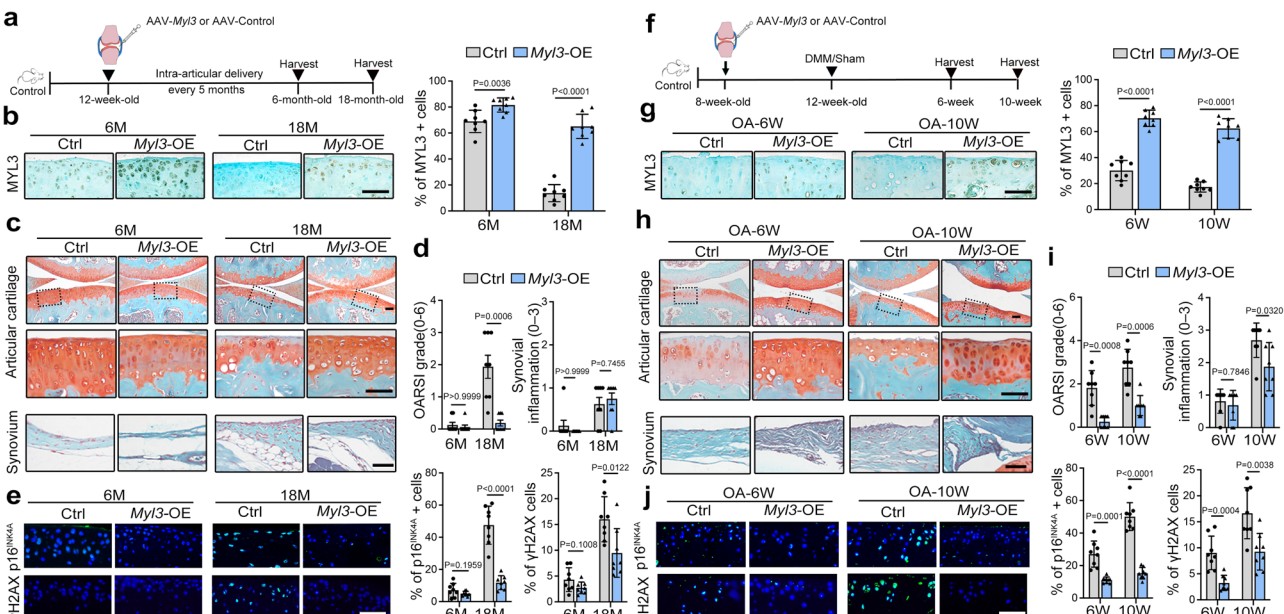

**Fig. 4 | Overexpression of MYL3 attenuates age-related and experimental OA development in mice. a** Schematic illustration of Ad-*Myl3* or Ad-control delivery schedules in an age-related OA model in mice. **b** Images and quantification of IHC staining of MYL3, Safranin-O staining images (**c**), OARSI and synovial inflammation scores (**d**), and IF staining images of p16^INK4A and γH2AX (**e**) in sagittal sections of joints from control and *Myl3*-OE mice at 6 months and 18 months. *n* = 8 independent biological replicates per group, bars = 50 μm. **f** Schematic illustration of Ad-*Myl3* or Ad-control delivery schedules in a posttraumatic OA model with DMM surgery in mice. **g** Images and quantification of IHC staining of MYL3, Safranin-O

staining images (**h**), OARSI and synovial inflammation scores (**i**), and IF staining images of p16^INK4A and γH2AX (**j**) in sagittal sections of joints from control and *Myl3*-OE mice at 6- and 10 weeks post surgery; *n* = 8 independent biological replicates per group, bars = 50 μm. Data are all shown as means ± SD. *P* values are from two-tailed Mann–Whitney *U* test (OARSI, synovial inflammation scores in **d**, **i**) and two-tailed unpaired *t* test (remaining quantification). *n* indicates the number of biologically independent samples or mice per group. Source data are provided as a Source Data file.

increased the levels of senescence markers at passages two and seven, while MYL3 overexpression or treatment with RO4929097 eliminated the enhanced senescence. Western blot analysis showed that expression of p16^INK4a, p53, γH2AX, HES1, and HEY1 were all upregulated in *Myl3*-KO chondrocytes but downregulated in *Myl3*-OE or *Myl3*-KO with inhibitor treatment (Supplementary Fig. 10a–d). Consistently, we found similar changes of SA-β-Gal activity in *Myl3*-KO, *Myl3*-OE, and *Myl3*-KO with inhibitor chondrocytes exposed to H2O2 (Supplementary Fig. 10e).

These data indicate that loss of MYL3 activates Notch signaling and induces senescence by enhancing CME, pharmacologic targeting of Notch signaling or inhibition of CME through pharmacological and genetic approaches prevents cellular senescence caused by MYL3 depletion in vitro.

### Inhibition of CME-Notch signaling prevents MYL3 loss-induced chondrocyte senescence and alleviates age-related and experimental OA

To confirm the critical role of targeting MYL3 and CME-Notch signaling in chondrocyte senescence, we compared the protective effect of the Notch pathway inhibitor RO4929097 and the CME inhibitor dynasore on the progression of posttraumatic or age-related OA in MYL3-KO and control mice. As expected, intraarticular injections of dynasore attenuated cartilage destruction in both control and *Myl3*-iKO mice at 6 weeks post-surgery. The severity of DMM-induced OA in *Myl3*-iKO mice was comparable to that in control mice after dynasore treatment (Fig. 6b, c, Supplementary Fig. 11a, b). Moreover, dynasore treatment significantly inhibited Notch signaling, eliminated senescent chondrocytes, and decreased SASP secretion in cartilages from both control and *Myl3*-iKO mice, as the expression of MMP13, NICD, HES1, p16^INK4a, and γH2AX were all downregulated and nuclear HMGB1 was upregulated (Fig. 6d and Supplementary Fig. 11c). Further, we used RO4929097 to treat *Myl3*-iKO and control mice after DMM surgery

(Fig. 6e). At 6 weeks post-surgery, mice treated with RO4929097 showed markedly reduced cartilage degeneration in comparison with vehicle treatment alone, as demonstrated by increased Safranin-O staining for proteoglycans, cartilage thickness and normalized OARSI scores. The severity of DMM-induced OA in *Myl3*-iKO mice was also comparable with that in control mice after RO4929097 treatment (Fig. 6f, g, Supplementary Fig. 12a). IHC staining confirmed the diminished numbers of senescent chondrocytes, decreased SASP factor and inhibited Notch signaling in articular cartilage of control and *Myl3*-iKO mice treated with RO4929097 (Fig. 6h, Supplementary Fig. 12b, c). Additionally, intraarticular injections of RO4929097 in both control and *Myl3*-KO mice at 18 months of age showed similar changes (Supplementary Fig. 13). Together, our data suggest that pharmacologic targeting of CME-Notch signaling in vivo would reduce the number of senescent chondrocytes and decrease SASP secretion, ultimately alleviating age-related and experimental OA.

## Discussion

Cellular senescence has emerged as a primary cause and an attractive therapeutic target of aging-associated chronic diseases, including OA[25,26]. However, how to define chondrocyte senescence and determine the underlying mechanisms remain key challenges for developing targeted therapy in OA[27]. With articular cartilage from aged mice, we screened and found that MYL3 expression was significantly reduced compared to young cartilage. In cultured chondrocytes and OA cartilage, we noticed that MYL3 expression negatively correlated with p21 and p16^INK4A, two major proteins that act as mediators of cell cycle arrest, which is associated with senescence. In vivo conditional deletion of *Myl3* in chondrocytes significantly promoted, whereas intraarticular injection of adeno-associated virus expressing MYL3 delayed, OA progression. Our data identify MYL3 as a negative indicator of chondrocyte senescence and OA progression.

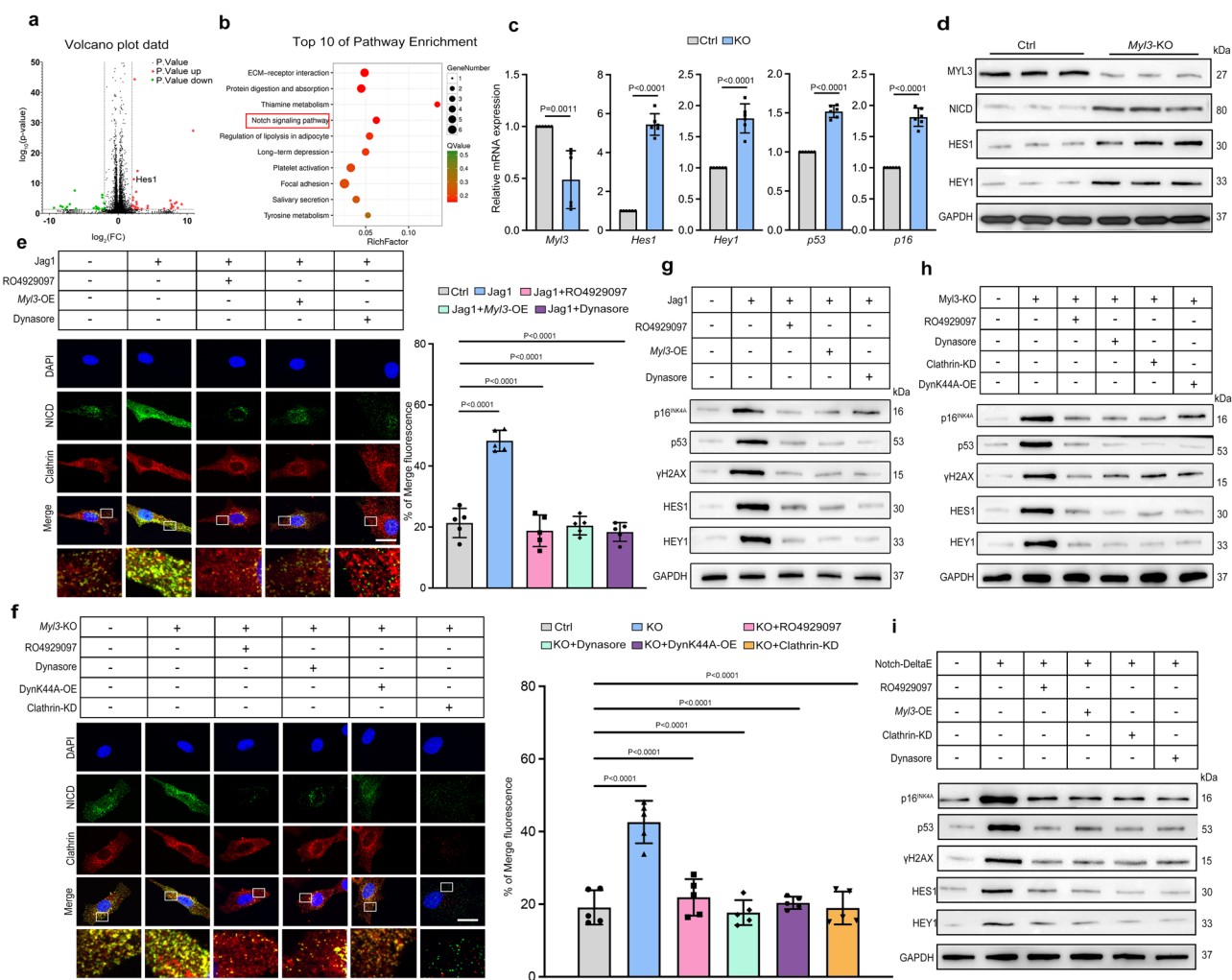

**Fig. 5 | Loss of MYL3 promotes the activation of Notch signaling and induces senescence by enhancing CME. a** Volcano plot of RNA-seq of primary chondrocytes from control and *Myl3*-KO mice. The dotted line denotes *P* = 0.05. **b** Top 10 pathway enrichment of KEGG pathway analysis following *Myl3* knockout in chondrocytes. **c** Relative mRNA expression of *Myl3, Hes1, Hey1, p16*[INK4A], and *p53*, and protein levels of MYL3, NICD, HES1, and HEY1 (**d**) in primary chondrocytes from control and *Myl3*-KO mice. **e** Images and quantification of IF staining for NICD (green) and clathrin (red) co-localization in mouse primary chondrocytes from control, jag1, jag1 + RO4929097, jag1+*Myl3*-OE, or jag1+dynasore group. *n* = 5, bars = 25 μm. **f** Images and quantification of IF staining for NICD (green) and cla-thrin (red) co-localization in primary chondrocytes from control, *Myl3*-KO, KO + RO4929097, KO+Dynasore, KO+DynK44A-OE, or KO+Clathrin-KD group. *n* = 5,

bars = 25 μm. **g** Protein levels of p16[INK4A], p53, γH2AX, HES1, and HEY1 in primary chondrocytes from control, jag1, jag1 + RO4929097, jag1+*Myl3*-OE, or jag1+dyna-sore group. **h** Protein levels of, p16[INK4A], p53, γH2AX, HES1, and HEY1 in primary chondrocytes from control, *Myl3*-KO, KO + RO4929097, KO+Dynasore, KO+DynK44A-OE, or KO+Clathrin-KD group. **i** Protein levels of p16[INK4A], p53, γH2AX, HES1, and HEY1 in primary chondrocytes from control, RO4929097, *Myl3*-OE, Clathrin-KD or Dynasore with vehicle or Notch-DeltaE. Data are representative of three independent experiments and are all shown as means ± SD. *P* values are from two-tailed unpaired *t* test (**c**) and one-way ANOVA test followed by Tukey's post hoc test (**e**, **f**). *n* indicates the number of biologically independent samples or mice per group. Source data are provided as a Source Data file.

Endocytosis is an essential cellular process for growth and survival in all eukaryotic cells, which mediates nutrient uptake, receptor internalization, and the regulation of cell signaling[28,29]. Aberrant endocytosis contributes to cellular senescence. CME and caveolae-mediated endocytosis represent major types of endocytosis that are implicated in senescence[30,31]. Previous studies claimed that senescent fibroblasts downregulate amphiphysin 1 among the CME components, coincident with reduced endocytosis of transferrin receptors[14]. How-ever, the elevation of transferrin expression was found in our study, indicating that CME was enhanced in senescent chondrocytes, which was in accordance with an early study that showed an increase in the rate and degree of endocytosis in aged chondrocytes[32]. More impor-tantly, reduced CME was previously demonstrated to be insufficient to induce senescence, as knockdown of clathrin adaptor complex AP2 causes growth arrest but does not recapitulate the senescent phenotype[15]. Mechanistically, we found that MYL3 negatively regulates

CME by inhibiting the interaction between MYO6 and clathrin. This study investigates the role of MYL3 in CME, and suggests that MYL3 may act as a negative mediator of CME in senescent cells. Our data reveals a previously unexplored molecular mechanism for enhanced CME, with loss of MYL3 promoting chondrocyte senescence in vivo and in vitro, and establishes the vital role of CME and MYL3 in the progression of OA.

To elucidate how increased CME caused by MYL3 deletion in chondrocytes promotes senescence, we carried out RNA sequencing and noted that the Notch pathway may be the most probable down-stream signaling pathway of MYL3 and CME in chondrocytes. Previous studies have proved that endocytosis plays an important role in the activation and regulation of Notch signaling[24,33,34], and activated Notch signaling triggers cellular senescence, characterized by increased p53 and p21 expression[35,36]. In our study, we found robust increased p21 expression and SA-β-gal activity in *Myl3* knockout chondrocytes, as

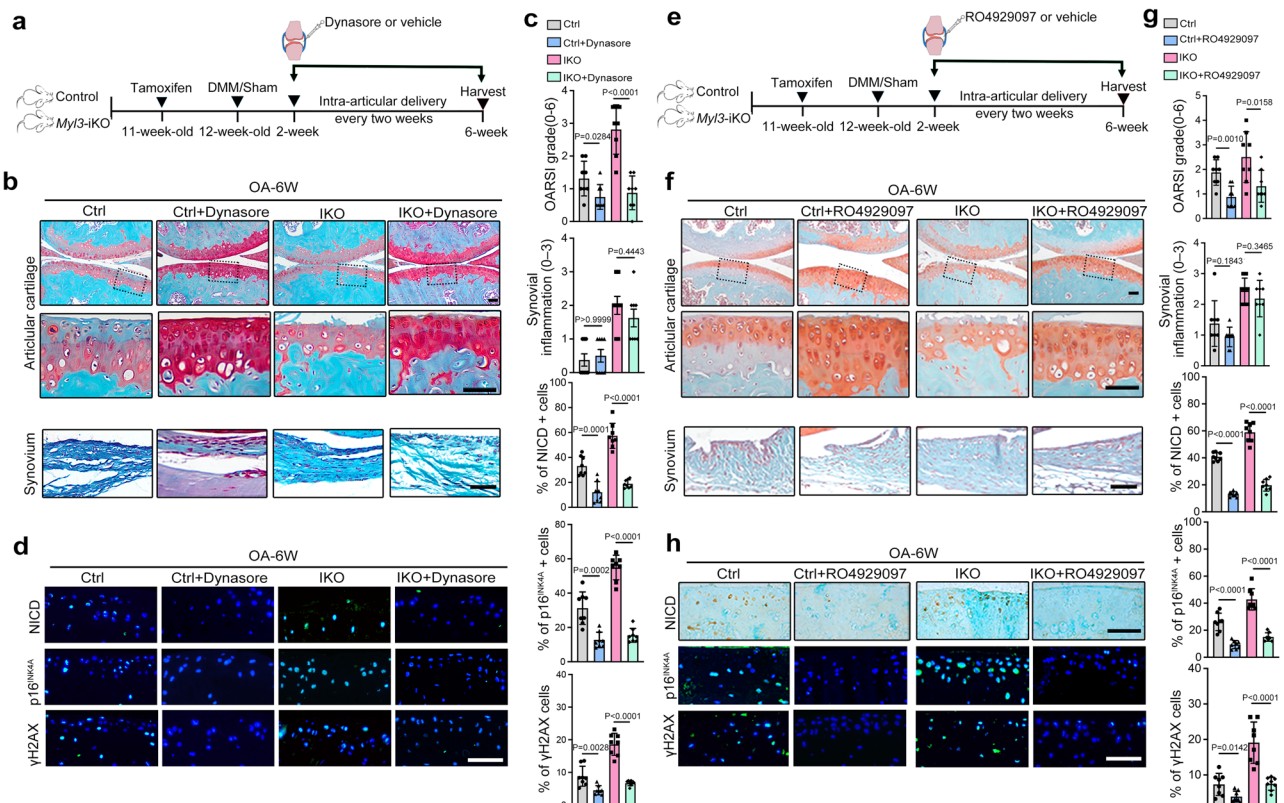

**Fig. 6 | Targeting MYL3-Notch signaling alleviates age-related and experimental OA. a** Schematic illustration of dynasore or PBS delivery schedules in a posttraumatic OA model with control and *Myl3*-iKO mice. **b** Safranin-O staining images and quantification of OARSI and synovial inflammation scores (**c**), images and quantification of IF staining for NICD, p16[INK4A], and γH2AX (**d**) in sagittal sections of joints from control or *Myl3*-iKO mice treated with dynasore or vehicle intraarticular injections at 6 weeks post surgery. *n* = 8 independent biological replicates per group, bars = 50 μm. **e** Schematic illustration of RO4929097 or vehicle delivery schedules in a posttraumatic OA model with control and *Myl3*-iKO mice. **f** Safranin-O staining images and quantification of OARSI and synovial

inflammation scores (**g**), images and quantification of IHC staining for NICD and IF staining for p16[INK4A] and γH2AX. **h** in sagittal sections of joints from control or *Myl3*-iKO mice treated with RO4929097 or vehicle intraarticular injections at 6 weeks post surgery. *n* = 8 independent biological replicates per group, bars = 50 μm. Data are all shown as means ± SD. *P* values are from two-tailed Mann–Whitney *U* test (synovial inflammation scores in **c**, **g**) and the two-tailed unpaired *t* test (remaining quantification). *n* indicates the number of biologically independent samples or mice per group. Source data are provided as a Source Data file.

well as enhanced CME of Notch, nuclear translocation of NICD, and expression of HES1.

These results are consistent with previous reports that Notch activation increases the transcriptional activity of p53, resulting in upregulation of p21 and premature senescence[37]. Overall, we conclude that MYL3 plays a negative role in CME-mediated Notch signaling activation and cellular senescence. Multiple studies have demonstrated that Notch signaling is a critical regulator of chondrogenesis and chondrocyte differentiation[38], but the exact role of Notch signaling in OA development remains controversial[39]. Sustained Notch activation in joint cartilage leads to early and progressive OA, while transient Notch activation results in enhanced synthesis of cartilage matrix and maintenance of cartilage[40,41]. Conditional deletion of RBPj or HES1 to inhibit Notch signaling in chondrocytes ameliorates experimental OA, but inhibition of Notch signaling by overexpression of a Notch1 antisense construct enhances chondrocyte hypertrophy and exacerbates experimental OA[42,43]. In our study, overexpression of *Myl3* or treatment with CME inhibitor and Notch signaling inhibitor RO4929097 to prevent the activation of Notch signaling in an aged or posttraumatic OA model resulted in alleviated chondrocyte senescence and cartilage destruction, suggesting that targeting overactivated Notch signaling is sufficient to prevent cellular senescence and OA development. Our study confirmed that activated Notch signaling is deleterious for chondrocyte homeostasis and would promote OA progression.

In summary, we conclude that during the development of OA, a decrease in the level of MYL3 in chondrocytes enhances the interaction of MYO6 with clathrin and increases CME of Notch receptor to activate Notch signaling, triggers chondrocytes senescence, promotes cartilage destruction, and aggravates OA progression. Supplementation of MYL3 and targeting blockade of CME-Notch signaling effectively ameliorates chondrocyte senescence and OA development in mice (Fig. 7). Our findings provide a molecular target for chondrocyte senescence and therapeutic strategies for OA. However, this study does have limitations. The precise dissection of trafficking dynamics involved in Notch activation following MYL3 downregulation remains elusive, given the variety of endocytosis/recycling subcellular compartments implicated in this complex process. Another limitation pertains to our inability to identify the specific cause for the reduced MYL3 protein level observed in senescent chondrocytes. Further investigation is warranted to accurately define the specific role of MYL3 in trafficking dynamics and to determine whether MYL3 impacts the senescence of cells other than chondrocytes.

## Methods
### Human cartilage information
Human OA cartilage was obtained from patients undergoing total knee replacement surgery in the Department of Orthopedic Surgery at the Third Affiliated Hospital of Southern Medical University (Guangzhou, China) (*n* = 8; aged 67 ± 2.03 years; five males and three females).

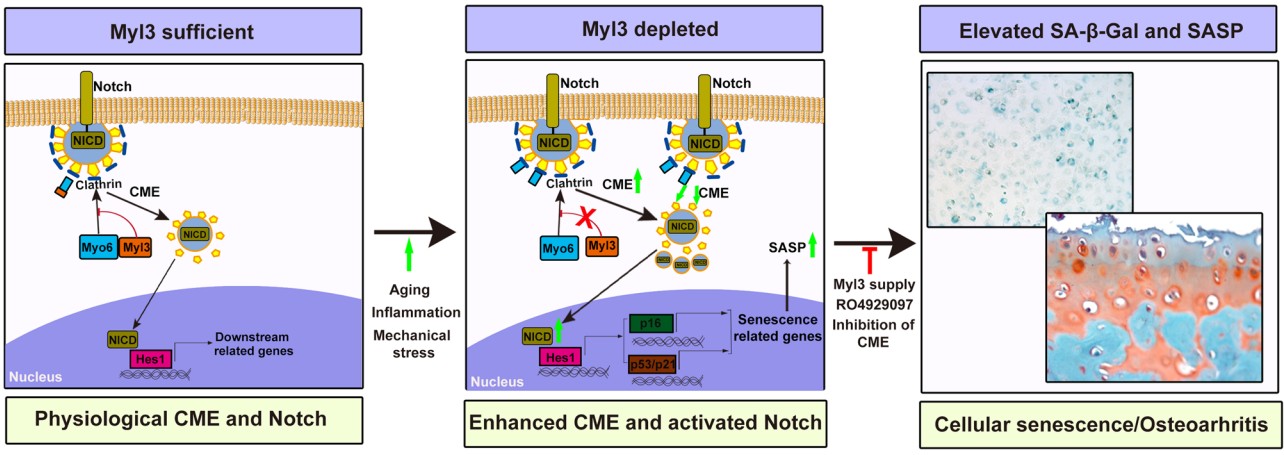

**Fig. 7 | Schematic of the role of MYL3 in mediating CME of Notch signaling and cellular senescence in chondrocytes.** In the process of aging or osteoarthritis development, a reduction of MYL3 in chondrocytes leads to enhanced CME by promoting the interaction between MYO6 and clathrin, and consequently induces the internalization of Notch receptors and nuclear translocation of NICD, resulting in the activation of Notch signaling. This process triggers chondrocytes senescence, promotes cartilage destruction, and aggravates OA progression. Supplementation of MYL3 and strategically blocking CME-Notch signaling effectively mitigates chondrocyte senescence and OA development in mice.

OA was confirmed in X-ray images of the knee-joint or in full-length X-ray images of both lower limbs. Control cartilage was collected from traffic incident victims with no history of arthritic disease ($n = 5$; aged $54 \pm 8.48$ years; three males and two females). Human cartilage samples were used for Safranin-O/fast green and Immunohistochemistry (IHC) staining. For human study, this project was approved by the Ethics Committee of the Third Affiliated Hospital of Southern Medical University. Written informed consent was obtained from all subjects before human tissue samples were harvested.

## Animals

All animal experiments were approved by the Animal Care and Use Committee of Southern Medical University (Guangzhou, China). Twelve-week-old male C57BL/6 J mice used for the experimental OA and 6-, 15-, and 18-month-old male C57BL/6 J mice used for aged-OA studies were purchased from the Laboratory Animal Center of Southern Medical University. For the *Myl3*-flox mice, an embryonic stem cell clone (Clone NO. HEPD0622_5_G02) was purchased from Cambridge-Suda (Cam-SU) Genomic Resource Center (Suzhou, China). Germline-transmitting chimeric mice were produced by Shanghai Model Organisms Center Inc. (Shanghai, China). The Col2a1-cre mouse line was a generous gift from Dr. Xiao Yang (Academy of Military Medical Sciences, Beijing, China), and *Col2a1-CreERT2* mice (Stock Number: 006774) were obtained from Jackson Laboratories (Bar Harbor, ME, USA). All mice were kept on the C57BL/6 J background and maintained in a standard, specific-pathogen-free facility of the Laboratory Animal Research Center of Southern Medical University. *Col2a1-Cre; Myl3*f ^lox/flox^ mice were designated *Myl3-KO, Col2a1-CreERT2; Myl3* ^flox/flox^ mice were designated *Myl3-iKO* and control littermates (*Myl3* ^flox/flox^) were referred to as controls. 12-week-old male transgenic mice were used for the experimental OA and 6-, 15-, and 18-month-old male transgenic mice were used for aged-OA studies. All C57BL/6 J and transgenic mice were housed in a specific specific-pathogen-free (SPF)-level animal room with five or fewer mice per cage. Mice had free access to food and water. All mice were maintained under circulating air, constant temperature (20–26 °C), and humidity of 50%–60% with a 12-hour light/dark cycle. *Myl3-iKO* mice and littermate control mice were intraperitoneally injected with 0.1 mg/g body weight tamoxifen (Sigma-Aldrich, CAS # 10540-29-1) dissolved in corn oil for 5 consecutive days at 11 weeks of age, and were subjected to DMM surgery at 12 weeks of age. Mice were euthanized under carbon dioxide or isoflurane (1.5–2% in $O_2$) for the collection of knee-joint

specimens at 6 or 10 weeks after DMM surgery or at 3 or 15 months after the first injection, and subjected to histological and biochemical analyses.

## Experimental OA model

12-week-old male C57BL/6 J mice, *Myl3*-iKO, and littermate control mice were subjected to surgical DMM surgery to induce OA. Briefly, the mice were anesthetized and their right knees were prepared for aseptic surgery. The joint capsule immediately medial to the patellar tendon was opened. The intercondylar region was exposed to visualize the meniscotibial ligament and this ligament was cut. The joint capsule and skin were then closed. In sham-treated animals, the joint capsule was opened but the meniscotibial ligament was not cut. The mice were euthanized under carbon dioxide or isoflurane (1.5–2% in $O_2$) for the collection of knee-joint specimens at 6 and 10 weeks after surgery.

## Intraarticular administration

Mice (C57BL/6 J, *Myl3* ^flox/flox^, *Myl3-KO,* and *Myl3-iKO*) were immobilized and the right leg was exposed for injection of 8 μL of Adeno-associated virus (AAV) expressing mouse MYL3 (AAV-*Myl3*) (Hanheng, Shanghai, China) or 20 μM RO4929097 (Selleck, S1575, China) into the joints using an insulin injection needle. AAV was first injected at 8 weeks of age and joints were harvested at 6 or 10 weeks after surgery or injected at 12 weeks of age and joints were harvested at 6 months or 18 months of age. RO4929097 was injected every two weeks starting at 2 weeks after DMM surgery and harvested at 6 or 10 weeks after surgery or injected every two weeks from the age of 12 weeks and harvested at 6 months or 18 months of age.

## Histological analysis

Human cartilage and mouse knee joints were fixed in 4% paraformaldehyde for 24 h, then decalcified in 10% EDTA (pH 7.4) for 21 days. The tissues were embedded in paraffin and sectioned continuously (3 μm thick) and serial sections were obtained from the medial and lateral compartments at 50 μm intervals. Seven to eight representative mid-sagittal sections were selected and deparaffinized in xylene, then hydrated with graded ethanol. Cartilage destruction was detected using Safranin-O/fast green staining. Slides were imaged using an Olympus-BX43 microscope (Olympus) with CellSens (v4.1) software. Degeneration of articular cartilage was quantified by three observers using the Osteoarthritis Research Society International (OARSI) scoring system under blinded conditions. Synovitis

inflammation was scored (grade 0–3) as described previously[44]. Osteophyte formation was identified by Safranin-O staining, and osteophyte size was measured with an Aperio Image Scope V12 (Leica Microsystems Ltd, Wetzlar, Germany).

## Cells

Rib cartilage and articular cartilage were isolated from newborn mice (24–72 h) under a stereo light microscope and digested with trypsin for 30 min. The primary chondrocytes obtained were purified and digested with 0.1% collagenase type II (Sigma-Aldrich, St Louis, MO, USA) with 10% FBS, 100 U/mL penicillin, and 100 mg/mL streptomycin sulfate for 4–6 h at 37 °C. The purified primary chondrocytes were resuspended, and then maintained as a monolayer in DMEM/F12 with 10% FBS, 100 U/mL penicillin, and 100 mg/mL streptomycin sulfate at 37 °C under 5% $CO_2$. The chondrogenic cell line ATDC5 was maintained in DMEM/F12 (Gibco/Life Technologies, Carlsbad, CA, USA) supplemented with 5% FBS, and 100 U/mL penicillin/streptomycin (Gibco/Life Technologies) at 37 °C under 5% $CO_2$ before subsequent treatment. Cells were infected with LV3-NC or LV3-*Myl3*-mus at 800 MOI, transfected with a *Myl3*-encoding plasmid using Lipofectamine 3000 (Invitrogen), treated with vehicle or $H_2O_2$ (200 µM) or treated with vehicle or RO4929097 (20 µM) for the indicated time periods.

## Transferrin endocytosis

Cells were incubated with cold live cell image solution (LCIS,140 mM NaCl, 20 mM Hepes, 2.5 mM KCl, 1.8 mM $CaCl_2$, and 1.0 mM $MgCl_2$, pH 7.4) containing 20 mM glucose and 1% BSA for 10 min on ice. Then the cells were incubated with Alexa Fluor 594–transferrin (20 µg/mL) in LCIS containing 20 mM glucose and 1% BSA for 2 min at 20 °C. After three washes with the corresponding media, cells were transferred back to 37 °C for the indicated times. Finally, cells were acid-washed before fixation to remove transferrin still bound at the cell surface. After fixation with 4% paraformaldehyde for 10 min, cells were observed using an Olympus FluoView confocal microscope (FV10, Olympus), and fluorescence intensity was analyzed with ImageJ software.

## Biotin-transferrin uptake assay

Cells were washed three times in Hank's balanced salt solution (pH 7.4) and then treated for 7 min at 37 °C with 8 µg/mL biotin-transferrin (Sigma-Aldrich) in the corresponding medium. After that, cells were washed three times with Hank's balanced salt solution (pH 4.0) to remove the remaining surface-bound biotin-transferrin. In order to determine the level of surface-bound transferrin, cells were incubated for 30 min with biotin-transferrin at 4 °C and washed three times with Hank's balanced salt solution (pH 7.4). Cell lysates were analyzed by immunoblotting.

## Statistics & reproducibility

Data are presented as means ± SD using SPSS version 20.0 software, and graphs were generated using GraphPad Prism 8.0. We used the Kolmogorov–Smirnov and Shapiro-Wilk test to assess data normality. The unpaired, two-tailed Student's *t* test or Mann-Whitney rank sum test (for non-normally distributed data) was used to compare two independent groups. For comparing multiple values, we used a one-way analysis of variance (ANOVA) with Tukey's post hoc test or Kruskal–Wallis test (for non-normally distributed data) with Dunn's multiple comparisons tests. All statistical tests used were two-sided. Values of $p < 0.05$ were considered statistically significant. Sample size for each experiment is indicated in the legend. Sample sizes were chosen to ensure adequate statistical power based on the size of effects observed and reaching statistical significance based on the available samples. All experiments were conducted on at least three independent biological replicates, including all histology and immunohistochemistry experiments. Western blot pictures are from a representative experiment and the number of independent repeats is clearly indicated in the figure legends. The number of biological replicates is indicated in figures. Throughout this text, "n" represents the number of independent biological replicates per group. For in vitro experiments, cell cultures are randomly assigned to each experimental group. Wild-type and transgenic mice were randomly assigned to each experimental group with various treatments. Randomization was performed when performing DMM surgeries on mice, with the surgeon unaware of the genotype of the mice. Littermate mice were ultimately compared based on genotype. All allocations were randomized in this study. For the mice surgeries and intraarticular injections, the operators were blinded to the group information. The investigators used sample ID and were not given grouping information during data collection. For histological analysis, the investigation was blinded to the group information, including genotype, treatment, or surgical condition of mice. WB, RT-PCR, IHC, and IF were performed by participants other than the experiment designer.

## LC/MS-MS mass spectrometry-based proteomics

Articular cartilage was carefully isolated from 2- and 12-month-old mice ($n = 3$ independent biological replicates per group) with microscissors under a stereoscopic microscope (Olympus, Tokyo, Japan), subchondral bones were removed and isolated articular cartilage was washed with PBS. Samples were lyophilized overnight and stored at −80 °C until shipment to the PGME Bio (Guangzhou, China) to perform LC–MS/MS.

The sample was prepared by grinding it into cell powder using liquid nitrogen. Four volumes of lysis buffer containing 1% SDS and 1% protease inhibitor cocktail were added to the cell powder and sonicated for three minutes on ice using a high-intensity ultrasonic processor. After sonication, the debris was removed by centrifugation at $12,000 \times g$ and 4 °C for 10 minutes. The protein concentration in the supernatant was determined using a BCA kit, according to the instructions provided by the manufacturer.

The protein sample encountered pre-cooled acetone treatment. Initially, one volume of acetone was used, followed by four volumes, resulting in precipitation at −20°C for a duration of 2 hours. The precipitate underwent 2–3 rounds of washing with pre-cooled acetone. Subsequently, the protein sample was dissolved in 200 mM TEAB and dispersed through the application of ultrasonication. Trypsin was introduced at a trypsin-to-protein mass ratio of 1:50 for the first digestion, which proceeded overnight. Afterward, the sample was subjected to reduction with 5 mM dithiothreitol at a temperature of 56 °C for 30 minutes, followed by alkylation with 11 mM iodoacetamide for 15 minutes in darkness at room temperature. Finally, peptide desalting was accomplished utilizing a Strata X SPE column.

The tryptic peptides were dissolved in solvent A and loaded onto a reversed-phase analytical column that was homemade. This column had dimensions of 25 cm in length and 100 µm in inner diameter. To separate the peptides, a mobile phase was used, consisting of solvent A (water containing 0.1% formic acid and 2% acetonitrile) and solvent B (water containing 0.1% formic acid and 90% acetonitrile). The separation protocol involved a gradient as follows: 0–68 min, 6%–23% B;68–82 min, 23%–32% B;82-86 min, 32%–80% B;86–90 min, 80% B. Subsequently, the separated peptides underwent analysis by an Orbitrap Exploris 480 equipped with a nano-electrospray ion source. For further MS/MS analyses, up to 25 precursors with the highest abundance were selected while implementing a dynamic exclusion of 20 seconds. The HCD fragmentation occurred at a normalized collision energy (NCE) of 27%. The automatic gain control (AGC) target was set to 100%, utilizing an intensity threshold of 50,000 ions/s and a maximum injection time of Auto.

The MaxQuant search engine (v.1.6.15.0) was utilized to process the MS/MS data and generate the desired results. To perform the search, the tandem mass spectra were compared against the

Mus_musculus_10090_SP_20230103.fasta database (with a total of 17,132 entries). To enhance the accuracy of the search, this database was combined with a reverse decoy and contaminants database. For the enzymatic cleavage, Trypsin/P was specified, allowing for a maximum of 2 missing cleavages. Additionally, the minimum peptide length was set to 7, and each peptide was allowed to have a maximum of 5 modifications. To ensure precision, a mass tolerance of 20 ppm was applied to the precursor ions in both the first and main search. Similarly, a mass tolerance of 20 ppm was applied to the fragment ions. In terms of modifications, the fixed modification was carbamidomethyl on Cys. On the other hand, the variable modifications included acetylation on the protein N-terminal and oxidation on Met. The false discovery rate for protein, peptide, and PSM was adjusted to be less than 1%.

The fold change (FC) was calculated as the ratio of the mean relative quantitative values of proteins in the two groups of samples. To determine the significance of the difference, a $T$ test was conducted on the relative quantitative values of protein in the two groups of samples, resulting in the calculation of a corresponding $P$ value, which served as the significance index. The default threshold for significance was set at a $P$ value < 0.05. To meet the normal distribution requirements of the $T$ test, the relative quantitative values of proteins were logarithmically converted using Log2 before the test. Through the above difference analysis, when $P$ value < 0.05, the change of differential expression level was more than 1.5 as the significantly upregulated change threshold, and 1/1.5 as the significantly downregulated change threshold.

## Immunohistochemistry (IHC) and IF

Specimens were prepared as described above and sections were incubated in citrate buffer (10 mM citric acid, pH 6.0) overnight at 60 °C to unmask the antigen. For IHC, 3% hydrogen peroxide was added for 10 min. Sections were blocked with 1% sheep serum for 1 h at 37 °C and incubated with primary antibodies (in 1% BSA, 0.1% Triton X-100) overnight at 4 °C. Subsequently, the sections were incubated with horseradish peroxidase-conjugated secondary antibodies (Jackson Immuno Research Laboratories, Inc., West Grove, PA, USA) for 1 h at room temperature. Finally, 3, 3-diaminobenzidine was used to observe the chromogen, with hematoxylin for counterstaining. All positively stained cells along the joint surface of each sample were counted in the femoral ankle and tibial plateau region, and the proportion of positive cells was evaluated using Image-Pro Plus 6.0 (Media Cybernetics, MD, USA). For IF staining, after incubation with primary antibodies, sections or cells seeded into a confocal dish were incubated with secondary antibodies conjugated with Alexa Fluor 488 or Alexa Fluor 594 (Life Technologies, Carlsbad, CA, USA) for 1 h at room temperature in the dark. Nuclei were labeled with 4, 6-diamidino-2-phenylindole (DAPI; Thermo Fisher Scientific, Waltham, MA, USA) and images were obtained using a FluoView FV1000 confocal microscope (Olympus) with FV10-ASW Viewer (v4.2) software. Images were analyzed using Image-Pro Plus (v6.0) software and Image J (v1.8). The number of positively stained cells in two fields per section of the articular cartilage and three sequential sections per mouse were counted in each group.

## SA-β-Gal staining

SA-β-Gal staining was performed using a Cell senescence β-galactosidase staining kit (Solarbio, G1580, Beijing, China) according to the manufacturer's protocol. Briefly, cells were washed with PBS and fixed for 15 min. Then, the cells were washed with PBS and incubated with SA-β-Gal staining solution for 16 h at 37 °C. SA-β-Gal-positive cells were calculated in three random fields per culture dish and analyzed using ImageJ (NIH, Bethesda, MD, USA) and senescence was scored as a percentage of SA-β-Gal-positive cells relative to the total cell number.

## Skeletal staining and histological analysis

Skeletons of whole-mount embryos were stained with Alcian blue and Alizarin red. In brief, whole embryos were skinned, eviscerated, fixed with 95% ethanol for 4 days, and immersed in acetone for 3 days. Samples were stained with one volume of 0.3% Alcian blue 8GX in 70% ethanol, one volume of 0.1% Alizarin red S in 95% ethanol, one volume of 100% acetic acid, and 17 volumes of 100% ethanol. The lengths of the resting/proliferative and hypertrophic zones were measured using a microscope image-analysis program (AxioVision).

## RNA sequencing

Total RNA was extracted using Trizol reagent (Invitrogen, Carlsbad, CA, USA) according to the manufacturer's protocol, and RNA quality was assessed on an Agilent 2100 Bioanalyzer (Agilent Technologies, Palo Alto, CA, USA) and checked using RNase-free agarose gel electrophoresis. Eukaryotic mRNA was enriched using Oligo beads and prokaryotic mRNA was enriched by removing rRNA with a Ribo-Zero™ Magnetic Kit (Epicentre, Madison, WI, USA). The enriched mRNA was fragmented into short fragments using fragmentation buffer and reverse transcribed into cDNA using random primers. Then second-strand cDNA was synthesized using DNA polymerase I, RNase H, dNTP, and buffer. The cDNA fragments were purified with a QiaQuick PCR extraction kit (Qiagen, Venlo, The Netherlands) to Illumina sequencing adapters. The ligation products were size selected by agarose gel electrophoresis, PCR amplified, and sequenced using an Illumina Novaseq 6000 by Gene Denovo Biotechnology Co. (Guangzhou, China). GO enrichment analysis was implemented using the GOseq R Package and DAVID online tool (https://david.ncifcrf.gov/). Pathways of differentially expressed genes were analyzed using the KEGG database (http://www.kegg.jp/kegg/) and GESA software (v4.0.3).

## RT-PCR analysis

RNA was isolated using Trizol (Invitrogen) and the concentrations were measured using a NanoDrop spectrophotometer (Thermo Fisher Scientific). Reverse transcription was performed with reverse transcriptase (Takara Bio, Shiga, Japan) according to the manufacturer's instructions. RT-PCR was performed using SYBR-green reagent (Yeasen Biotechnology, Shanghai, China) with the appropriate primers according to the manufacturer's instructions. Quantitative polymerase chain reaction was performed using LightCycler® 96 (Roche) with LightCycler® 96 software 1.1. PCR thermocycling conditions were as follows: 95 °C for 2 min, 94 °C for 15 s, 60 °C for 15 s, and 72 °C for 20 s, for a total of 40 cycles. Comparative quantification was performed using the 2-ΔΔCq method. The expression of each gene was normalized to the level of the housekeeping gene GAPDH. RT-PCR data was analyzed in Microsoft Excel version 16.36 and graphs were generated using GraphPad Prism 8.0. The primer sequences used in this study were: *GAPDH* forward 5'-TGG CCT TCC GTG TTC CTA C-3' and reverse 5'-GAG TTG CTG TTG AAG TCG CA-3'; *Myl3* forward 5'-TGC CTC CAA GAT TAA GAT CGA GT-3' and reverse 5'-CTC TGC CTG GGT AGG ATT CTG-3'; *Hes1* forward 5'- GAT AGC TCC CGG CAT CCC AAG-3' and reverse 5'- GCG CGG TAT TTC CCC AAC A-3'; *Hey1* forward 5'- TGG CAG AAG TTG CGC GTT AT-3' and reverse 5'-CGC TGG GAA GCG TAG TTG TT-3; *p16* forward 5'-GCT CAA CTA CGG TGC AGA TTC-3' and reverse 5'-GCA CGA TGT CTT GAT GTC CC-3'; *p53* forward 5'-CCC CTG TCA TCT TTT GTC CCT-3' and reverse 5'-AGC TGG CAG AAT AGC TTA TTG AG-3'.

## Co-immunoprecipitation assay

For the co-immunoprecipitation assay, cells were washed twice with ice-cold PBS and then lysed in lysis buffer (40 mM HEPES [pH 7.4], 2 mM EDTA, 10 mM pyrophosphate, 10 mM glycerophosphate, 0.3% CHAPS and one tablet of EDTA-free protease inhibitors (Roche, Basel,

Switzerland) per 25 mL) on ice for 30 min. After centrifugation for 10 min at 12,000 × g and 4 °C, primary antibodies were added to the supernatant and rotated overnight at 4 °C. A 50% slurry of Protein A + G Sepharose (Santa Cruz Biotechnology, Santa Cruz, CA, USA) was then added to the samples, and the samples were rotated for 1 h at 4 °C. The immunoprecipitates were washed three times with lysis buffer, the pellets were dissolved in 2× SDS loading buffer after centrifugation and boiled for 10 min at 100 °C. Proteins were analyzed by immunoblotting with the indicated antibodies.

## Immunoblotting
Tissues and cells were lysed in 2× SDS buffer (62.5 mM Tris-HCl [pH 6.8], 10% glycerol, 2% SDS, 50 mM dithiothreitol, 0.01% bromophenol blue) on ice and heated for 10 min at 96 °C. The lysates were then separated using 10% sodium dodecyl sulfate-polyacrylamide gel electrophoresis and transferred to nitrocellulose membranes (Bio-Rad Laboratories, Berkeley, CA, USA) by the wet transfer method. Each membrane was blocked with 5% nonfat milk in 100 mM Tris-HCl pH 7.5, 150 mM NaCl, 0.05% Tween 20, for 1 h at room temperature, and then incubated with primary antibodies overnight at 4 °C on a shaker. The following day the membranes were incubated with an appropriate horseradish peroxidase-conjugated secondary antibody for 1 h at room temperature. The reacted proteins were detected with enhanced chemiluminescence reagents (Santa Cruz Biotechnology). Tanon image (v1.0) software and GeneSys (v1.8.5) used for imaging and data acquisition.

## Antibodies
The following antibodies were used for IHC or IF analysis: peroxidase-conjugated affinipure goat anti-mouse IgG (1:500, 115-035-003, Jackson ImmunoResearch, USA), peroxidase-conjugated affiniPure goat anti-rabbit IgG (1:500, 111-035-003, Jackson ImmunoResearch), anti-rabbit Alexa Fluor™ 488 (1:400, a11008, Invitrogen, USA), anti-rabbit Alexa Fluor™ 594 (1:400, a11012, Invitrogen), anti-mouse Alexa Fluor™ 488 (1:400, a21202, Invitrogen), anti-mouse Alexa Fluor™ 594 (1:400, a21203, Invitrogen), mouse monoclonal anti-MYL3 (1:200, MLM527, ab680; Abcam, Cambridge, UK), rabbit polyclonal anti-MYL3 (1:100, 10913-1-AP, Proteintech, Rosemont, IL, USA), mouse monoclonal anti-p16 (1:200, 2D9A12, ab54210, Abcam), rabbit polyclonal anti-p16 (1:200, A0262, Abclonal, Wuhan, CN), rabbit monoclonal anti-γH2AX (1:200, EP8542Y, ab81299, Abcam), rabbit polyclonal anti-HMGB1 (1:200, 10829-1-A, Proteintech), rabbit polyclonal anti-MMP13 (1:200, A1606, Abclonal), rabbit polyclonal anti-NICD (1:200, ab8925, Abcam), rabbit polyclonal anti-MYO6(1:200, 26778-1-AP, Proteintech), rabbit polyclonal anti-MYO6(1:200, ab230478, Abcam), rabbit polyclonal anti-clathrin (1:200, ab21679, Abcam), mouse monoclonal anti-clathrin (1:200, X22, ab2731, Abcam), mouse monoclonal anti-RAB5 (1:200, 3A4, ab66746, Abcam), mouse monoclonal anti-RAB7 (1:200, Rab7-117, ab50533, Abcam), rabbit monoclonal anti-HES1 (1:100, ARC0513, A0925, Abclonal). The following antibodies were used for western blotting: peroxidase-conjugated affinipure goat anti-mouse IgG (1:5000, 115-035-003, Jackson ImmunoResearch, USA), peroxidase-conjugated affinipure goat anti-rabbit IgG (1:5000, 111-035-003, Jackson ImmunoResearch), rabbit polyclonal anti-MYL3 (1:1000, 10913-1-AP, Proteintech), rabbit polyclonal anti-p16 (1:1000, A0262, Abclonal), rabbit polyclonal anti-MMP13 (1:2000, A1606, Abclonal), rabbit polyclonal anti-p21 (1:2000, A1483, Abclonal), mouse monoclonal anti-p53 (1:2000, Clone 1C12, 2524 S, Cell Signaling Technology, Danvers, USA), rabbit monoclonal anti-γH2AX (1:1000, EP8542Y, ab81299, Abcam), rabbit polyclonal anti-MYO6 (1:1000, 26778-1-AP, Proteintech), rabbit polyclonal anti-MYO6 (1:1000, ab230478, Abcam), rabbit polyclonal anti-Clathrin (1:1000, ab21679, Abcam), rabbit polyclonal anti-HEY1 (1:1000, A16110, Proteintech) and rabbit monoclonal anti-HES1 (1:1000, A0925, Abclonal).

## Reporting summary
Further information on research design is available in the Nature Portfolio Reporting Summary linked to this article.

## Data availability
The original RNA-seq data generated in this study have been deposited in the GEO database under accession code GSE232325. The mass spectrometry proteomics data have been deposited to the ProteomeXchange Consortium via the PRIDE partner repository with the dataset identifier PXD044684. All other relevant data supporting the findings of this study are available within the article and its Supplementary Information file. Source data are provided with this paper.

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

## Acknowledgements

This work was supported by grants from the National Natural Science Foundation of China (Grant no. 31900840 for K.L., 82172499 for K.L., 81871745 for A.L., 32160209 for J.L.), President Foundation of The Third Affiliated Hospital of Southern Medical University (Grant no. 2.0004 for K.L.), Guangxi Key Laboratory of basic and translational research of Bone and Joint Degenerative Diseases (21-220-06 for Y.T.). The authors thank Le Hu, Shuqin Zhang, and Wenping Chen for providing secretarial assistance and technical support.

## Author contributions

Study design: K.L. and X.B. Data analysis: H.C. Data collection: H.C., P.Y., J.L., H.L., P.L., H.W., A.L., and Y.S. Drafting manuscript: H.C., P.Y., Y.T., and B.G. Approving final version of manuscript: all authors. H.C., P.Y., and J.L. contributed equally to this work. K.L. and X.B. take responsibility for the integrity of the data analysis.

## Competing interests

The authors declare no competing interests.
