## [Peer Review file · Nature Communications]

REVIEWER COMMENTS

Reviewer #1 (Remarks to the Author):

This work concerns very relevant and interesting aspects in cartilage tissue physiology and pathology, related to the mechanisms responsible for chondrocyte senescence and development of OA disease. In particular, through the integration of in vitro and in vivo models highly relevant to the study of chondrocyte and cartilage tissue physiopathology with proteome-wide analyses, the Authors identify the myosin light chain (Myl3) protein as a major regulator of the cartilage tissue homeostasis, whose loss correlates with enhanced expression of senescence-associated phenotypes in chondrocytes and OA progression. The Authors also provide biochemical and functional evidence that, by physically interacting with Myosin VI (Myo6) and inhibiting Myo6-clathrin association, Myl3 negatively regulated clathrin-mediated endocytosis (CME). Furthermore, by integrating transcriptomic data obtained in Myl3-KO primary chondrocytes with functional assays in model systems, the Authors ultimately propose a model in which Myl3 deficiency results in enhanced CME-dependent internalization and ensuing activation of Notch signaling in chondrocytes, associated with chondrocyte senescence and enhanced OA progression.

Based on these premises, this study has the potential to highlight important aspects related to the debated and apparently dual opposite role of Notch signaling in normal cartilage homeostasis and OA pathogenesis, with important implications for the therapy of cartilage degenerative/inflammatory disease. There are, however, a number of issues, described in detail thereafter, that need to be conceptually and experimentally addressed. In general terms, while the association of Myl3 deficiency with the acquisition of senescence phenotypes in vitro and accelerated OA progression in vivo is elegantly demonstrated through patho-physiologically relevant mouse models and validated in human samples, evidence that Myl3 negatively controls Notch signaling through CME remains at this stage indirect and correlative, which heavily detracts from the enthusiasm for this study in its present form.

There are a number of experimental approaches and conceptual issues that should be thoroughly addressed and discussed to raise the overall scientific impact and novelty of this study.

Major points:

The study starts with the observation that, following serial passages in cultures, chondrocytes express senescent markers apparently associated with increased CME, as assessed by the use of a fluorochrome-conjugated transferrin. It would really strengthen the functional significance of this findings, which represents the founding observation of the study, if the Authors could demonstrate that CME activation is causal to the de novo appearance of senescent features. To this aim, the Authors could leverage the broad and well-established array of genetic and/or pharmacological tools to achieve transient/reversible as well as prolonged inhibition of CME. It is advisable that these tools are used in the context of time-course experiments performed in the described cell-based models, for instance in serially passaged as well as H2O2-treated chondrocytes. The authors should prioritize on the following experiments:

- Targeted inhibition of CME through pharmacological (for instance, using the reversible dynamin inhibitor, dynasore) and genetic (clathrin silencing and enforced DynK44A mutant expression) approaches should be performed to assess the actual impact of enhanced CME activation on the stepwise appearance of senescent markers in a time-dependent fashion.
- The above experimental tools should be used in parallel experiments to assess the impact of CME on the development and progression of OA in vivo (intra-articular injection with adenovirus expressing DynK44A or CME-inhibitory drugs, similar to the experiments performed with adeno-Myl3), and to test the actual dependency of aberrant Notch activation by increased CME in the different model systems in vitro and in vivo, for instance including experiments based on Myl3 depletion in the DMM surgery

model. It would be also interesting to see the effects of intra-articular Notch inhibitors in wild type DMM surgery-treated mice, given the purported role of Notch in the cartilage repair process. In this regard, an immunohistochemistry analysis to show intranuclear Notch accumulation in the damaged vs intact cartilage and a time-course analysis during the repair vs inflammatory evolution process would also be informative.

- The demonstration that CME is enhanced upon Myl3 loss in the DMM model as well in OA patients should also be provided.

- While biochemical and immunofluorescence data arguing for a direct role of Myl3 in regulating CME by interfering with clathrin/Myo6 interaction are convincing, evidence of the existence of a direct mechanistic link between this Myl3 endocytic function and the downstream effects on senescence and OA progression remain correlative in the absence of phenotype reversion experiments based on the above described pharmacological and genetic tools. In this context, it would also be paramount to perform time-course immunofluorescence experiments using inducible silencing of Myl3 to follow the very early steps of endocytosis and perform a thorough characterization of the different early vs late endosomal compartment (including clathrin itself, as well rab5/7) that have been reported to be involved in CME-dependent Notch activation (see also a similar point below). Indeed, after prolonged CME activation, cells might have enacted so many possible compensatory events, mostly relying on activation of the trafficking/recycling pathway, which might affect interpretation and confound the actual significance of the results.

- Evidence that Notch signaling is activated via CME in senescent chondrocytes relies solely on co-localization experiments of NICD with clathrin, upon treatment with a jag1 recombinant protein or H2O2 exposure. One might argue that this enhanced association is the downstream consequence of Notch activation caused by mechanisms occurring in the context of the overall inflammation process that might be caused by the absence of Myl3, rather than being specifically linked to enhanced CME activation. Indeed, in the proteome-wide analysis of the OA patients, desmin and actinin-2, which are intimately involved at various steps in the regulation of endocytosis and also linked to Notch signaling, emerged among the top downregulated genes alongside Myl3. Further complicating this scenario, it should also be considered the possibility that enhanced CME, downstream of Myl3 deficiency, might also affect Notch ligands rather than the Notch receptor per se. Related to these points, it should be noted that the predominant view of the regulatory role of endocytosis in Notch signaling activation is that, in the canonical ligand-dependent pathway, endocytosis of Notch ligands in the signal-sending cell is required to generate the pulling force on the extracellular domain of Notch expressed on the signal-receiving cell, where this event is prerequisite to the conformational changes associated with the consecutive proteolytic cleavages exerted by ADAMs and presenilin-gamma-secretase, which eventually lead to the intranuclear translocation of the Notch intracellular domain. In contrast, the need for internalization of Notch itself in the signal-receiving cell as a prerequisite for signal transduction remains controversial. While several lines of evidence involve early steps of the endocytic process in the suppression of Notch signaling, blocking the endocytic trafficking at later stages enhances Notch signaling, likely as a consequence of the prolonged retention of Notch in an environment where it can be processed by presenilin-gamma-secretase. Furthermore, ligand-independent activation of Notch is described as a consequence of ion and calcium concentration alterations and/or intracellular acidification, which might typically occur in inflammatory processes.

- It would be interesting to assess whether a constitutively active form of Notch, retaining the site for gamma-secretase inhibition (namely the extracellular domain-deprived Notch mutant, Notch-DeltaE) would force a senescence program that could be reversed by gamma-secretase inhibitors and, even more relevant to the overall conceptual frame of this work, by CME pharmacological and genetic inhibition.

Additional considerations related to individual figures/experiments:

Fig1/S1.

In the immunofluorescence comparison of control vs. H₂O₂-treated or P2 vs. P7 chondrocytes, it seems that the basal steady state transferrin internalization rate is extremely, if any, low in unstressed cells. Related to this, the reported ~3-fold difference in the quantitative analysis of these experiments appears not to reflect the dramatic differences depicted in the immunofluorescence images. It would be useful to perform these experiments including an additional early endocytosis/trafficking marker, such as rodamine EGF. It is also paramount in this type of experiments to quantitatively assess the steady state number of transferrin receptors present on the cell surface (for instance, with biotin-based experiments) to guard against the possible influence of altered surface receptor expression on the internalization rate. Time-course experiments are paramount to rule out the intervention of compensatory mechanisms due to receptor recycling activation, combined with the use of markers for the different subcellular endosomal compartments.

Another major problem is that, very often, different time points are considered in the different experimental settings, in particular for senescence and OA, which makes difficult to internally compare the results. A case in point is the proteome-wide analysis performed comparing 2- vs. 12-month old mice (Fig.1B); the expression analysis of Myl3 in tissues has been done at 3, 10 and 18 months (Fig1C). Increasing the consistency across the different experiments, at least for the most relevant experimental settings would certainly aid a better understanding of the results.

Fig. S1b and S1c.

Three different lanes are shown for each comparison. Do they represent three independent preparations of untreated and H₂O₂-treated cells (in this case it would be concerning the heterogeneous behavior of some markers, p21 and Myl3 in b and p53 in c), or do they represent different time-points or treatment concentrations? There are no details regarding this point in the related figure legend (please note that the same problem is in Fig. 5d).

It is also not clear in the analysis of this figure why the mouse ATDC5 cell line was introduced in this set of experiments, given the availability of mouse primary chondrocytes. While this increases the heterogeneity of the model systems, it detracts from the highly patho-physiologically relevant model represented by the primary chondrocytes. Data on p53, shown for the ATDC5 cell line should therefore be obtained for primary chondrocytes as well. Ideally, these cells could also be used for the immunofluorescence experiments depicted in Fig. S1d.

Fig. 2e and S2a,c,d.

The set of results from immunofluorescence studies on Myo6/clathrin co-localization in Myl3KD or Myl3-OE cells is really difficult to interpret based on the rather poor quality resolution of the images. Apparently, the bulk of this interaction occurs in what apparently look like as Golgi-associated structures or in any case with structures far from being in close proximity to the plasma membrane, such as CCP or early endosomes, as one would expect from early CME-associated endosomal structures. As stated above, time-course experiments, combined with high-resolution microscopy approaches, or proximity ligation assay experiments combined with appropriate early vs late endosomal markers would be instrumental to shedding lights on these fundamental questions. It is also important to include in these experiments the staining for Myl3 in either KD or OE cells, with the aim to precisely perform the colocalization analysis exclusively in silenced or overexpressing cells, reporting their relative percentage in the quantitation analysis.

Fig. S2b.

Myo6/clathrin co-immunoprecipitation upon H₂O₂ exposure should be also performed in conditions of Myl3KD and Myl3-OE.

Fig. 2.

Considering the importance of the WT vs. Myl3-KO genetic model for the connection between enhanced CME/senescence/OA and perturbed Myl3 levels, it becomes paramount to introduce a control for the CME activation state (for instance transferrin internalization at different time-points) in the comparison of WT vs. Myl3-KO and Myl3-KO+Myl3-OE cells, and including stemming results in Fig. 2 o in Fig. S2.

Fig. 4 and S4.

To corroborate the existence of a connection between Myl3 deficiency, senescence and OA progression, it is fundamental to show basally increased CME in DMM and OA mice. Of note, it would be appropriate to describe the functional setting of the DMM surgery (and specify the acronym as well) for the general reader.

Fig. 5c.

Additional Notch targets should be included in the qPCR analysis (for instance, other Hes or Hey family targets), considering that Hes1 might be downstream of other signaling pathways (activated especially in tissue degenerative or inflammatory conditions). These targets should be used in experiments with Notch inhibitors.

Fig. 5e,h.

This figure suffers from the same problem described for Fig. 2 in terms of extremely poor quality of images and intrinsic difficulty to interpret the nature of the endosomal/intracellular structures. Please refer to the considerations/suggestions made for the Fig. 2 as concerns the use of early vs. late endosomal markers and time-course experiments.

What do the squared boxes represent in d? Do they point to a clathrin/NICD subcellular association confined to the perinuclear region (which would be puzzling per se and deserve detailed explanation)? Notch inhibitors should be included as controls in these experiments and, most importantly, genetic silencing of clathrin as well as DynK44A overexpression.

Reviewer #2 (Remarks to the Author):

In this manuscript, the authors demonstrate that chondrocyte senescence in mice is regulated by clathrin-mediated endocytosis (CME). With aging, they find decreased Myl3 which enhances CME and results in increased Notch activation. The latter then leads to accelerated chondrocyte senescence.

Overall, these are a logical series of experiments that are of interest and significance in terms of the development of OA. My main concern is their incomplete linkage, especially *in vivo*, of their findings to senescence.

First, in the initial study, they compare proteome expression in rapidly growing, 2 month old mice, versus 12 month old mice. They then link this to senescence. But here they are really comparing rapidly growing 2 month old mice to middle aged mice. So the changes they observed could have nothing to do with senescence, as senescent cells really do not begin to accumulate in larger numbers in mice until much later (after at least 18 months of age). How do they know that the changes they are observing are not just related to completion of growth and development? For example, in Fig 1d, 10 and 18 months were different from 3 months, but 10 months was no different from 18 months.

Further compounding this problem is that for all of the *in vivo* work, their findings are limited to showing increased p16Ink4a and MMP13 expression. In the Methods, I cannot find any information on what p16 antibody they used or how they validated it. To claim effects on senescence *in vivo*, they need more than this. Specifically, provide evidence of telomeric DNA damage (eg, TAFs, SADs, or other assays). Without this, they really cannot make claims regarding *in vivo* senescence effects.

An additional point is why only male mice were used in the studies? Are the findings not present in female mice? In general, NIH now requires studying both sexes, or justifying not doing so.

Reviewer #3 (Remarks to the Author):

This is potentially an interesting article, suggesting a new proposed mechanism of chondrocyte ageing/senescence, that may have a role in the progression of OA.

I have several concerns regarding the proteomic methodologies described:

- In general I found it extremely difficult throughout to ascertain the biological cf. technical replicates used.
- The proteomic techniques used are very poorly described, what was the strategy for protein labelling? Pooling of samples? Validation? Why were only n=1 human samples used, indeed the n=3 comparison of mice seems far too low. The FC cut-offs used need justification. Were there 2x cut-offs applied? One for absolute analyte differences and one for pathway analyses? The Bioinformatics analysis is not described.

Response

Reviewer #1:

This work concerns very relevant and interesting aspects in cartilage tissue physiology and pathology, related to the mechanisms responsible for chondrocyte senescence and development of OA disease. In particular, through the integration of in vitro and in vivo models highly relevant to the study of chondrocyte and cartilage tissue physiopathology with proteome-wide analyses, the Authors identify the myosin light chain (My13) protein as a major regulator of the cartilage tissue homeostasis, whose loss correlates with enhanced expression of senescence-associated phenotypes in chondrocytes and OA progression. The Authors also provide biochemical and functional evidence that, by physically interacting with Myosin VI (Myo6) and inhibiting Myo6-clathrin association, My13 negatively regulated clathrin-mediated endocytosis (CME). Furthermore, by integrating transcriptomic data obtained in My13-KO primary chondrocytes with functional assays in model systems, the Authors ultimately propose a model in which My13 deficiency results in enhanced CME-dependent internalization and ensuing activation of Notch signaling in chondrocytes, associated with chondrocyte senescence and enhanced OA progression.

Based on these premises, this study has the potential to highlight important aspects related to the debated and apparently dual opposite role of Notch signaling in normal cartilage homeostasis and OA pathogenesis, with important implications for the therapy of cartilage degenerative/inflammatory disease. There are, however, a number of issues, described in detail thereafter, that need to be conceptually and experimentally addressed. In general terms, while the association of My13 deficiency with the acquisition of senescence phenotypes in vitro and accelerated OA progression in vivo is elegantly demonstrated through patho-physiologically relevant mouse models and validated in human samples, evidence that My13 negatively controls Notch signaling through CME remains at this stage indirect and correlative, which heavily detracts from the enthusiasm for this study in its present form.

There are a number of experimental approaches and conceptual issues that should be thoroughly addressed and discussed to raise the overall scientific impact and novelty of this study.

Major points:

The study starts with the observation that, following serial passages in cultures, chondrocytes express senescent markers apparently associated with increased CME, as assessed by the use of a fluorochrome-conjugated transferrin. It would really strengthen the functional significance of this findings,

which represents the founding observation of the study, if the Authors could demonstrate that CME activation is causal to the de novo appearance of senescent features. To this aim, the Authors could leverage the broad and well-established array of genetic and/or pharmacological tools to achieve transient/reversible as well as prolonged inhibition of CME. It is advisable that these tools are used in the context of time-course experiments performed in the described cell-based models, for instance in serially passaged as well as H2O2-treated chondrocytes. The authors should prioritize on the following experiments:

- Targeted inhibition of CME through pharmacological (for instance, using the reversible dynamin inhibitor, dynasore) and genetic (clathrin silencing and enforced DynK44A mutant expression) approaches should be performed to assess the actual impact of enhanced CME activation on the stepwise appearance of senescent markers in a time-dependent fashion.

Response: Thank you for this valuable comment. According to your suggestion, we evaluated the protein levels of p16^{INK4A}, p53, and γ H2AX in primary chondrocytes from control, dynasore, DynK44A-OE, or clathrin-KD at passage 2 or 7 (Supplementary Figure 2b–c). Our results demonstrated that pharmacological or genetic inhibition of CME decreased the levels of senescence markers at passages two and seven.

Supplementary Figure 2

- The above experimental tools should be used in parallel experiments to assess the impact of CME on the development and progression of OA *in vivo* (intra-articular injection with adenovirus expressing DynK44A or CME-inhibitory drugs, similar to the experiments performed with adeno-Myl3), and to test the actual dependency of aberrant Notch activation by increased CME in the different model systems *in vitro* and *in vivo*, for instance including experiments based on Myl3 depletion in the DMM surgery model. It would be also interesting to see the effects of intra-articular Notch inhibitors in wild type DMM surgery-treated mice, given the purported role of Notch in the cartilage repair process. In this regard, an immunohistochemistry analysis to show intranuclear Notch accumulation in the damaged vs intact cartilage and a time-course analysis during the repair vs inflammatory evolution process would also be informative.

Response: Thank you very much for this comment. It is true that the role of CME in OA progress is very important in our study. As suggested, we compared the protective effect of the dynamin inhibitor dynasore on the progression of posttraumatic OA in Myl3-iKO and control mice (Figure 6a–d, Supplementary Figure 11). Our data suggested that targeting CME using pharmacologic method would reduce the numbers of senescent cells and decrease SASP secretion *in vivo*. The effects of intra-articular Notch inhibitors in wild type mice underwent DMM surgery have already been shown in our study (Figure 6e–h, Supplementary Figure 12-13). In the revised manuscript, we have added IF analysis to show intranuclear Notch accumulation in the damaged vs intact cartilage and posttraumatic or age-related OA (Supplementary Figure 7a-c).

Figure 6

Supplementary Figure 11

Supplementary Figure 12

Supplementary Figure 13

Supplementary Figure 7

- The demonstration that CME is enhanced upon Myl3 loss in the DMM model as well in OA patients should also be provided.

Response: We have analyzed the co-localization of Myo6 and clathrin levels in the OA model as well in OA patients (Supplementary Figure 1d, e and f), and showed the enhancement of co-localization of clathrin and NICD levels in cartilage from Myl3-KO mice (Supplementary Figure 7e).

Supplementary Figure 1

Supplementary Figure 7

- While biochemical and immunofluorescence data arguing for a direct role of Myl3 in regulating CME by interfering with clathrin/Myo6 interaction are convincing, evidence of the existence of a direct mechanistic link between this Myl3 endocytic function and the downstream effects on senescence and OA progression

remain correlative in the absence of phenotype reversion experiments based on the above described pharmacological and genetic tools. In this context, it would also be paramount to perform time-course immunofluorescence experiments using inducible silencing of Myl3 to follow the very early steps of endocytosis and perform a thorough characterization of the different early vs late endosomal compartment (including clathrin itself, as well rab5/7) that have been reported to be involved in CME-dependent Notch activation (see also a similar point below). Indeed, after prolonged CME activation, cells might have enacted so many possible compensatory events, mostly relying on activation of the trafficking/recycling pathway, which might affect interpretation and confound the actual significance of the results.

Response: We performed the time-course experiments to evaluate the role of Myl3 in regulating CME and found that Myl3-knockdown lentivirus (Myl3-KD) had a stronger transferrin uptake than control, while overexpression of Myl3 showed less (Figure 2a). In addition, consistent with the results from experiments using H₂O₂ stimulation, knockdown of Myl3 enhanced, while overexpression of Myl3 inhibited, transferrin uptake (Supplementary Figure 3a). We also added time-course experiments and found that Myl3 loss induced enhancement of CME could be blocked by dynasore, DynK44A overexpression and clathrin knockdown (Supplementary Figure 9c), and the enhanced expression and co-localization of NICD with Clathrin, Rab5 or Rab7 caused by jag1 treatment could be blocked by treatment with the Notch pathway inhibitor RO4929097, dynasore and Myl3 overexpression (Figure 5e, Supplementary Figure 8). We also found that compared with control, Myl3-KO chondrocytes had elevated expression level of NICD and enhanced co-localization of NICD with Clathrin, Rab5 or Rab7. And the above observations could be eliminated by RO4929097, dynasore, DynK44A overexpression, or clathrin knockdown (Figure 5f, Supplementary Figure 9a-b).

Figure 2

Supplementary Figure 3

Figure 5

e

Jag1	-	+	+	+	+
RO4929097	-	-	+	-	-
MyI3-OE	-	-	-	+	-
Dynasore	-	-	-	-	+

● Ctrl ▲ Jag1 ■ Jag1+RO4929097
◆ Jag1+MyI3-OE ● Jag1+Dynasore

f

MyI3-KO	-	+	+	+	+	+
RO4929097	-	-	+	-	-	-
Dynasore	-	-	-	+	-	-
Clathrin-KD	-	-	-	-	+	-
DynK44A-OE	-	-	-	-	-	+

● Ctrl ▲ KO ■ KO+RO4929097
◆ KO+Dynasore ● KO+Clathrin-KD ▼ KO+DynK44A-OE

Supplementary Figure 8

Supplementary Figure 9

- Evidence that Notch signaling is activated via CME in senescent chondrocytes relies solely on co-localization experiments of NICD with clathrin, upon treatment with a jag1 recombinant protein or H₂O₂ exposure. One might argue that this enhanced association is the downstream consequence of Notch activation caused by mechanisms occurring in the context of the overall inflammation process that might be caused by the absence of Myl3, rather than being specifically linked to enhanced CME activation. Indeed, in the proteome-wide analysis of the OA patients, desmin and actinin-2, which are intimately involved at various steps in the regulation of endocytosis and also linked to Notch signaling, emerged among the top downregulated genes alongside Myl3. Further complicating this scenario, it should also be considered the possibility that enhanced CME, downstream of Myl3 deficiency, might also affect Notch ligands rather than the Notch receptor per se. Related to these points, it should be noted that the predominant view of the regulatory role of endocytosis in Notch signaling activation is that, in the canonical ligand-dependent pathway, endocytosis of Notch ligands in the signal-sending cell is required to generate the pulling force on the extracellular domain of Notch expressed on the signal-receiving cell, where this event is prerequisite to the conformational changes associated with the consecutive proteolytic cleavages exerted by ADAMs and presenilin-gamma-secretase, which eventually lead to the intranuclear translocation of the Notch intracellular domain. In contrast, the need for internalization of Notch itself in the signal-receiving cell as a prerequisite for signal transduction remains controversial. While several lines of evidence involve early steps of the endocytic process in the suppression of Notch signaling, blocking the endocytic trafficking at later stages enhances Notch signaling, likely as a consequence of the prolonged retention of Notch in an environment where it can be processed by presenilin-gamma-secretase. Furthermore, ligand-independent activation of Notch is described as a consequence of ion and calcium concentration alterations and/or intracellular acidification, which might typically occur in inflammatory processes.

Response: Thank you for this valuable comment. According to your suggestion, we have investigated whether the Notch-activation induced by Myl3-loss could be blocked by CME inhibition and the Notch-activation caused by Jag1 could be blocked by CME inhibition and Myl3 overexpression. We found that the upregulation of p16INK4a, p53, γ H2AX, Hes1, and Hey1 caused by Myl3 knockout could be eliminated by dynasore, DynK44A overexpression, or clathrin knockdown (Figure 5h). In addition, western blot analysis showed that the expression levels of p16INK4a, p53, γ H2AX, Hes1, and Hey1 were all upregulated with jag1 treatment but downregulated in jag1 with RO4929097, dynasore, or Myl3 overexpression treatment (Figure 5g). The enhancement in co-localization of NICD with Clathrin, Rab5, or

Rab7 caused by jag1 could be blocked by treatment with the Notch pathway inhibitor RO4929097, dynasore, or Myl3 overexpression (Figure 5e, Supplementary Figure 8). The similar Myl3 loss-enhanced co-localization of NICD with Clathrin, Rab5, or Rab7 could be eliminated by RO4929097, dynasore, DynK44A overexpression, or clathrin knockdown (Figure 5f, Supplementary Figure 9a-b). These data indicated that the Myl3-loss induced Notch activation could be blocked by CME inhibition supported that Notch activation caused by the absence of Myl3 is linked to enhanced CME activation.

Figure 5

- It would be interesting to assess whether a constitutively active form of Notch, retaining the site for gamma-secretase inhibition (namely the extracellular domain-deprived Notch mutant, Notch-DeltaE) would force a senescence program that could be reversed by gamma-secretase inhibitors and, even more relevant to the overall conceptual frame of this work, by CME pharmacological and genetic inhibition.

Additional considerations related to individual figures/experiments:

Response: We have shown that Notch-DeltaE could upregulate the expression of p16INK4a, p53, γH2AX, Hes1, and Hey1, and could be reversed by RO4929097, Myl3 overexpression, dynasore, and clathrin knockdown (Figure 5i).

Figure 5

Fig1/S1.

In the immunofluorescence comparison of control vs. H₂O₂-treated or P2 vs. P7 chondrocytes, it seems that the basal steady state transferrin internalization rate is extremely, if any, low in unstressed cells. Related to this, the reported ~3-fold difference in the quantitative analysis of these experiments appears not to reflect the dramatic differences depicted in the immunofluorescence images. It would be useful to perform these experiments including an additional early endocytosis/trafficking marker, such as rodamine EGF. It is also paramount in this type of experiments to quantitatively assess the steady state number of transferrin receptors present on the cell surface (for instance, with biotin-based experiments) to guard against the possible influence of altered surface receptor expression on the internalization rate. Time-course experiments are paramount to rule out the intervention of compensatory mechanisms due to receptor recycling activation, combined with the use of markers for the different subcellular endosomal compartments.

Response: Thank you for this valuable comment. We have performed time-course experiments to show the changes of transferrin internalization rate in primary chondrocytes at passage seven or with H₂O₂ treatment (Figure 1b and Supplementary Figure 1b). And we also performed biotin-based experiments to show the upregulated transferrin internalization rate in primary chondrocytes at passage seven or with H₂O₂ treatment (Supplementary Figure 1c). We have tried to perform time-course experiments combined with the markers for the different subcellular endosomal compartments. However, we did not get valid visual data. We have noticed the importance of these experiments and we will keep trying to perform the experiments to get valid data in our future study.

Figure 1

Supplementary Figure 1

Another major problem is that, very often, different time points are considered in the different experimental settings, in particular for senescence and OA, which makes difficult to internally compare the results. A case in point is the proteome-wide analysis performed comparing 2- vs. 12-month old mice (Fig.1B); the

expression analysis of Myl3 in tissues has been done at 3, 10 and 18 months (Fig1C). Increasing the consistency across the different experiments, at least for the most relevant experimental settings would certainly aid a better understanding of the results.

Response: Thank you for this comment. We re-analyzed Myl3 expression in cartilages from mice at 2, 12 and 24 months (Figure 1d) to increase the consistency.

Figure 1

Fig. S1b and S1c.

Three different lanes are shown for each comparison. Do they represent three independent preparations of untreated and H₂O₂-treated cells (in this case it would be concerning the heterogeneous behavior of some markers, p21 and Myl3 in b and p53 in c), or do they represent different time-points or treatment concentrations? There are no details regarding this point in the related figure legend (please note that the same problem is in Fig. 5d).

It is also not clear in the analysis of this figure why the mouse ATDC5 cell line was introduced in this set of experiments, given the availability of mouse primary chondrocytes. While this increases the heterogeneity of the model systems, it detracts from the highly patho-physiologically relevant model represented by the primary chondrocytes. Data on p53, shown for the ATDC5 cell line should therefore be

obtained for primary chondrocytes as well. Ideally, these cells could also be used for the immunofluorescence experiments depicted in Fig. S1d

Response: The three lanes are biological repeats on the same condition which described in the “Materials and Methods” section. We used ATDC5 because this cell line is commonly used in the field of chondrocyte senescence as a supplement for primary chondrocytes. We fully agree that the use of ATDC5 can increase the heterogeneity of the model systems. Thus, we have deleted these data and focused our study on primary chondrocytes.

Fig. 2e and S2a, c, d.

The set of results from immunofluorescence studies on Myo6/clathrin co-localization in Myl3KD or Myl3-OE cells is really difficult to interpret based on the rather poor-quality resolution of the images. Apparently, the bulk of this interaction occurs in what apparently look like as Golgi-associated structures or in any case with structures far from being in close proximity to the plasma membrane, such as CCP or early endosomes, as one would expect from early CME-associated endosomal structures. As stated above, time-course experiments, combined with high-resolution microscopy approaches, or proximity ligation assay experiments combined with appropriate early vs late endosomal markers would be instrumental to shedding lights on these fundamental questions. It is also important to include in these experiments the staining for Myl3 in either KD or OE cells, with the aim to precisely perform the colocalization analysis exclusively in silenced or overexpressing cells, reporting their relative percentage in the quantitation analysis.

Response: Thank you for these valuable comments. We have performed the time-course experiments and showed the Myl3 knockdown lentivirus (Myl3-KD) lead to a stronger transferrin uptake than control, while overexpression of Myl3 showed less in primary chondrocytes (Figure 2a). We improved the quality of immunofluorescence images on Myo6/clathrin co-localization to show the changes near the plasma membrane, as we expected from early CME-associated endosomal structures (Figure 2e, Supplementary Figure 3c).

Figure 2

Supplementary Figure 3

Fig. S2b.

Myo6/clathrin co-immunoprecipitation upon H₂O₂ exposure should be also performed in conditions of Myl3KD and Myl3-OE.

Response: We have performed the co-immunoprecipitation of Myo6 and clathrin in primary chondrocytes with vehicle or H₂O₂ treatment in conditions of Myl3-KD and Myl3-OE (Supplementary Figure 3b).

Supplementary Figure 3

b

Fig. 2.

Considering the importance of the WT vs. Myl3-KO genetic model for the connection between enhanced CME/senescence/OA and perturbed Myl3 levels, it becomes paramount to introduce a control for the CME activation state (for instance transferrin internalization at different time-points) in the comparison of WT vs. Myl3-KO and Myl3-KO+Myl3-OE cells, and including stemming results in Fig. 2 o in Fig. S2.

Response: We have performed transferrin internalization at different time points in Myl3-KO+Myl3-OE cells (Figure 2a, Supplementary Figure 3a) and have introduced a control group for the CME activation state in Myl3-KO cells to show the change of transferrin internalization (Supplementary Figure 9c).

Figure 2

Supplementary Figure 3

Supplementary Figure 9

Fig. 4 and S4.

To corroborate the existence of a connection between Myl3 deficiency, senescence and OA progression, it is fundamental to show basally increased CME in DMM and OA mice. Of note, it would be appropriate to describe the functional setting of the DMM surgery (and specify the acronym as well) for the general reader.

Response: Thank you for this comment. We have analyzed the co-localization levels of Myo6 and clathrin in OA mice and OA patients (Supplementary Figure 1d, e and f) to show increased CME. We also added the description of the functional settings about the surgical destabilization of the medial meniscus (DMM) surgery and specify the acronym in the part of experimental OA model.

Supplementary Figure 1

Fig. 5c.

Additional Notch targets should be included in the qPCR analysis (for instance, other Hes or Hey family targets), considering that Hes1 might be downstream of other signaling pathways (activated especially in tissue degenerative or inflammatory conditions). These targets should be used in experiments with Notch inhibitors.

Response: We have added hey1 in qPCR analysis (Figure 5c) and western blot analysis in experiments with Notch inhibitors (Figure 5d and g-i, Supplementary Figure 10b, d).

Figure 5

c

d

g

h

i

Supplementary Figure 10

b

d

Fig. 5e, h.

This figure suffers from the same problem described for Fig. 2 in terms of extremely poor quality of images and intrinsic difficulty to interpret the nature of the endosomal/intracellular structures. Please refer to the considerations/suggestions made for the Fig. 2 as concerns the use of early vs. late endosomal markers and time-course experiments.

What do the squared boxes represent in d? Do they point to a clathrin/NICD subcellular association confined to the perinuclear region (which would be puzzling per se and deserve detailed explanation)? Notch inhibitors should be included as controls in these experiments and, most importantly, genetic silencing of clathrin as well as DynK44A overexpression.

Response: We have performed the time-course experiments to show the Myl3-loss induced CME activation could be blocked by the inhibition of CME (Supplementary Figure 9c). We have improved the quality of immunofluorescence images on NICD/clathrin co-localization. (Figure 5e and f).

We found that the enhanced expression and co-localization of NICD with Clathrin, Rab5, or Rab7 caused by jag1 treatment could be blocked by treatment with the Notch pathway inhibitor RO4929097, dynasore, or Myl3 overexpression (Figure 5e, Supplementary Figure 8). Expression of NICD and its co-localization with Clathrin, Rab5, or Rab7 were enhanced in Myl3-KO chondrocytes, and could be eliminated by RO4929097, dynasore, DynK44A overexpression and clathrin knockdown (Figure 5f, Supplementary Figure 9a-b).

Figure 5

e

Jag1	-	+	+	+	+
RO4929097	-	-	+	-	-
MyI3-OE	-	-	-	+	-
Dynasore	-	-	-	-	+

● Ctrl ▲ Jag1 ■ Jag1+RO4929097
◆ Jag1+MyI3-OE ● Jag1+Dynasore

f

MyI3-KO	-	+	+	+	+	+
RO4929097	-	-	+	-	-	-
Dynasore	-	-	-	+	-	-
Clathrin-KD	-	-	-	-	+	-
DynK44A-OE	-	-	-	-	-	+

● Ctrl ▲ KO ■ KO+RO4929097
◆ KO+Dynasore ● KO+Clathrin-KD ▼ KO+DynK44A-OE

Supplementary Figure 8

Supplementary Figure 9

Reviewer #2:

In this manuscript, the authors demonstrate that chondrocyte senescence in mice is regulated by clathrin-mediated endocytosis (CME). With aging, they find decreased Myl3 which enhances CME and results in increased Notch activation. The latter then leads to accelerated chondrocyte senescence.

Overall, these are a logical series of experiments that are of interest and significance in terms of the development of OA. My main concern is their incomplete linkage, especially *in vivo*, of their findings to senescence.

First, in the initial study, they compare proteome expression in rapidly growing, 2 month old mice, versus 12 month old mice. They then link this to senescence. But here they are really comparing rapidly growing 2 month old mice to middle aged mice. So the changes they observed could have nothing to do with senescence, as senescent cells really do not begin to accumulate in larger numbers in mice until much later (after at least 18 months of age). How do they know that the changes they are observing are not just related to completion of growth and development? For example, in Fig 1d, 10 and 18 months were different from 3 months, but 10 months was no different from 18 months.

Response: Thank you for your valuable comment. We used this time point because other studies in this field had reported chondrocyte senescence at 12-month-old^{1, 2, 3, 4}. While noticing the argument about the time point, we performed our aged-related OA model at 18 months of age. Also, we have changed the time points to 2, 12 and 24 months to analysis the expression of Myl3 in tissues and to detect the chondrocyte senescence by IF staining of p16INK4A γ H2AX, and HMGB1 (Figure 1d, Supplementary Figure 1d).

Figure 1

Supplementary Figure 1

Further compounding this problem is that for all of the *in vivo* work, their findings are limited to showing increased p16Ink4a and MMP13 expression. In the Methods, I cannot find any information on what p16 antibody they used or how they validated it. To claim effects on senescence *in vivo*, they need more than this. Specifically, provide evidence of telomeric DNA damage (eg, TAFs, SADs, or other assays). Without this, they really cannot make claims regarding *in vivo* senescence effects.

Response: Thank you for your comment. The information on p16 antibody has been described in the part of Antibody in Supplementary Materials and Methods. We now added γH2AX to show the DNA damage in our study both *in vitro* and *in vivo* and based on the similar study in chondrocyte senescence in age related OA⁵. We also added HMGB1, whose nuclear expression precedes the secretion of SASP components in

cells undergoing senescence, to assess the progressing chondrocyte senescence. (Figure 1-6, Supplementary Figure 1-3, 5-6, 10-13).

Figure 1 Supplementary Figure 1

Figure 4 Supplementary Figure 6

Figure 3 Supplementary Figure 5

Figure 6 Supplementary Figure 11 Supplementary Figure 12

Supplementary Figure 13

Figure 2

Figure 5

Supplementary Figure 1

Supplementary Figure 2

Supplementary Figure 3

Supplementary Figure 10

An additional point is why only male mice were used in the studies? Are the findings not present in female mice? In general, NIH now requires studying both sexes, or justifying not doing so.

Response: Thank you for your comment. Gender is a very important factor in age-related OA and chondrocyte senescence. The spontaneous osteoporosis which occurs in old female mice could affect the subchondral bone, which is also an important characteristic in OA pathology. To avoid this, we use male mice in our study, according other similar studies in chondrocyte senescence^{6,7 8}.

Reviewer #3:

This is potentially an interesting article, suggesting a new proposed mechanism of chondrocyte ageing/senescence, that may have a role in the progression of OA.

I have several concerns regarding the proteomic methodologies described:

- In general I found it extremely difficult throughout to ascertain the biological cf. technical replicates used.

Response: Thank you for pointing this out. We have added descriptions on biological and technical replicates in the revised figure legends.

- The proteomic techniques used are very poorly described, what was the strategy for protein labelling? Pooling of samples? Validation? Why were only n=1 human samples used, indeed the n=3 comparison of mice seems far too low. The FC cut-offs used need justification. Were there 2x cut-offs applied? One for absolute analyte differences and one for pathway analyses? The Bioinformatics analysis is not described.

Response: Thank you for pointing this out. The proteomic techniques used here have been added in “Supplementary Materials and Methods” section. In this study, we used isobaric tag for relative and absolute quantitation (iTRAQ) labeling LC-MS/MS proteomics analysis to identify proteins with altered expression level in cartilages from young vs aged mice, and intact vs damage cartilages from OA patient. We did not perform pathway analyses with our proteomic data. Total protein was extracted from human and mice samples pool, with n=1 human and n=3 mice used, as we only want to identify all the changing proteins. The protein peptides were labelled with 8-plex iTRAQ reagents performed using an iTRAQ reagent Multiplex kit (Applied Biosystems, Foster City, CA, USA) according to the manufacturer’s protocol. The used cut-offs were described as: “Differentially expressed proteins were filtered for a fold change cutoff of 1.5 or 0.5 and a p-value cutoff of 0.05.” For bioinformatics analysis, GO enrichment analysis was implemented using the Goseq R Package and DAVID online tool (<https://david.ncifcrf.gov/>). Pathways of differentially expressed genes were analyzed by the KEGG database

(<http://www.kegg.jp/kegg/>) and GESA software. We have added the missing information in the revised “Supplementary Materials and Methods” section.

References

1. Akoum J, *et al.* Aging Cartilage in Wild-Type Mice: An Observational Study. *Cartilage* **13**, 1407S-1411S (2021).
2. Chen J, *et al.* 1,25-Dihydroxyvitamin D Deficiency Accelerates Aging-related Osteoarthritis via Downregulation of Sirt1 in Mice. *International journal of biological sciences* **19**, 610-624 (2023).
3. Li L, *et al.* Positive Effects of a Young Systemic Environment and High Growth Differentiation Factor 11 Levels on Chondrocyte Proliferation and Cartilage Matrix Synthesis in Old Mice. *Arthritis & rheumatology* **72**, 1123-1133 (2020).
4. Rowe MA, *et al.* Reduced Osteoarthritis Severity in Aged Mice With Deletion of Macrophage Migration Inhibitory Factor. *Arthritis & rheumatology* **69**, 352-361 (2017).
5. Jeon OH, *et al.* Local clearance of senescent cells attenuates the development of post-traumatic osteoarthritis and creates a pro-regenerative environment. *Nature medicine* **23**, 775-781 (2017).
6. Loeser RF, Kelley KL, Armstrong A, Collins JA, Diekman BO, Carlson CS. Deletion of JNK Enhances Senescence in Joint Tissues and Increases the Severity of Age-Related Osteoarthritis in Mice. *Arthritis & rheumatology* **72**, 1679-1688 (2020).
7. Deng L, *et al.* Stabilizing heterochromatin by DGCR8 alleviates senescence and osteoarthritis. *Nat Commun* **10**, 3329 (2019).
8. Xie J, *et al.* Sustained Akt signaling in articular chondrocytes causes osteoarthritis via oxidative stress-induced senescence in mice. *Bone Research* **7**, (2019).

REVIEWERS' COMMENTS

Reviewer #1 (Remarks to the Author):

A major point of criticism raised during the initial round of revision of this manuscript concerned the indirect and correlative nature of findings supporting evidence that Myl3 exerts a negative control over Notch signaling through CME, and that loss of this functional axis was the underlying molecular working responsible for the acquisition of senescence phenotypes in chondrocytes and ensuing degenerative events correlated to OA progression. Consequently, this Reviewer suggested quite a broad array of in vitro and in vivo experiments, based on integrated pharmacological and genetic approaches, deemed to be necessary to strengthen the physiopathological relevance of the original study. Following these indications, the Authors have made quite a remarkable amount of additional experimental work to address all the major questions and concerns raised in the first round of revision, which have enabled them to craft a much more improved version of their study. The only major point that remains unaddressed regards the exact definition of the trafficking kinetic events throughout the endosomal compartment, which are hypothesized to underpin Notch activation following Myl3 downregulation and ensuing CME activation. While this would have certainly represented a mechanistic aspect of the outmost biological relevance, the lack of its precise definition does not heavily detract from the overall impact of the study, provided that the Authors clearly state, for instance in the Discussion section, that the precise dissection of the trafficking dynamics involved in Notch activation downstream of Myl3 downregulation remains elusive in this study, as is the involvement of the different endocytosis/recycling subcellular compartments.

Reviewer #2 (Remarks to the Author):

The authors have performed additional analyses, including those more specific for senescence, and added an additional aged mice cohort. They have satisfactorily addressed my concerns.

Reviewer #3 (Remarks to the Author):

I have been asked to comment specifically on the validity of the proteomics used. Unfortunately, I don't think the technique has been used appropriately. The cut-offs used need some further justification and the very low n numbers are worrying. I would consider removing this part of the study if no further work can be undertaken to strengthen the results.

Response

Reviewer #1:

A major point of criticism raised during the initial round of revision of this manuscript concerned the indirect and correlative nature of findings supporting evidence that Myl3 exerts a negative control over Notch signaling through CME, and that loss of this functional axis was the underlying molecular working responsible for the acquisition of senescence phenotypes in chondrocytes and ensuing degenerative events correlated to OA progression. Consequently, this Reviewer suggested quite a broad array of in vitro and in vivo experiments, based on integrated pharmacological and genetic approaches, deemed to be necessary to strengthen the physiopathological relevance of the original study. Following these indications, the Authors have made quite a remarkable amount of additional experimental work to address all the major questions and concerns raised in the first round of revision, which have enabled them to craft a much more improved version of their study. The only major point that remains unaddressed regards the exact definition of the trafficking kinetic events throughout the endosomal compartment, which are hypothesized to underpin Notch activation following Myl3 downregulation and ensuing CME activation. While this would have certainly represented a mechanistic aspect of the outmost biological relevance, the lack of its precise definition does not heavily detract from the overall impact of the study, provided that the Authors clearly state, for instance in the Discussion section, that the precise dissection of the trafficking dynamics involved in Notch activation downstream of Myl3 downregulation remains elusive in this study, as is the involvement of the different endocytosis/recycling subcellular compartments.

Response: We sincerely appreciate your insightful comments. Regarding the remaining major point, we have added a clear statement in our Discussion section acknowledging that the exact dissection of trafficking dynamics involved in Notch activation following Myl3 downregulation remains unclear in this study, as does the involvement of various endocytosis/recycling subcellular compartments. Building on the existing foundation and the guidance you provided during this study, we plan to delve deeper into these mechanisms in our future research.

During the initial round of manuscript revision, a significant critique was raised regarding the indirect and correlative nature of findings that suggested Myl3 exerts negative control over Notch signaling through CME. To address these concerns, we have conducted further experiments, incorporating both pharmacological and genetic methodologies in vitro and in vivo, based on your valuable and insightful comments. The extensive

additional experimental work was deemed essential to bolster the pathophysiological relevance of our study, thus enabling us to present a more refined version. We extend our heartfelt gratitude for your invaluable insights that greatly assisted our research process.

Reviewer #2:

The authors have performed additional analyses, including those more specific for senescence, and added an additional aged mice cohort. They have satisfactorily addressed my concerns.

Response: We express our gratitude for your insightful comments. Your suggestions have indeed broadened our understanding of senescence considerably. By incorporating these additional analyses based on your comments, including more specific elements for senescence and an extra cohort of aged mice in our study, we have enriched the quality of our research under your guidance.

Reviewer #3:

I have been asked to comment specifically on the validity of the proteomics used. Unfortunately, I don't think the technique has been used appropriately. The cut-offs used need some further justification and the very low n numbers are worrying. I would consider removing this part of the study if no further work can be undertaken to strengthen the results.

Response: We are grateful for your comments and extend our apologies for the inappropriate use of proteomics. Following your suggestion, we have repeated the proteomics experiment utilizing mouse cartilage samples and provided sufficient details of the experiment in the Methods section. In the revised manuscript, we have added the current result of the analysis based on the repeated proteomics experiment (Figure 1c). Even though changes in expression levels of the most significantly altered proteins and those within the Myosin family differ from our previous outcomes (Figure 1c, Supplementary Table 1), a consistent tendency towards MYL3 down-regulation is still evident, preserving its position as the most significantly suppressed member of the Myosin family in our study (Figure 1c.) Unfortunately, due to difficulty in obtaining sufficient human samples within a limited timeframe, we were unable to repeat the proteomics experiment with human samples. Consequently, taking your advice into consideration, we have chosen to remove this portion of our data. We will ensure to avoid such oversights in our future study, guided by your valuable comments.